



# The wind farm as a sensor: learning and explaining orographic and plant-induced flow heterogeneities from operational data

Robert Braunbehrens[1], Andreas Vad[1], and Carlo L. Bottasso[1]

[1]Wind Energy Institute, Technische Universität München, D-85748 Garching b. München, Germany

**Correspondence:** Carlo L. Bottasso (carlo.bottasso@tum.de)

**Abstract.** This paper describes a method to identify the heterogenous flow characteristics that develop within a wind farm in its interaction with the atmospheric boundary layer. The whole farm is used as a distributed sensor, which gauges through its wind turbines the flow field developing within its boundaries. The proposed method is based on augmenting an engineering wake model with an unknown correction field, which results in a hybrid (grey-box) model. Operational SCADA data is then used to simultaneously learn the parameters that describe the correction field, and tune the ones of the engineering wake model. The resulting monolithic maximum likelihood estimation is in general ill-conditioned because of collinearity and low observability of the redundant parameters. This problem is solved by a singular value decomposition, which discards parameter combinations that are not identifiable given the informational content of the dataset, and solves only for the identifiable ones.

The farm-as-a-sensor approach is demonstrated on two wind plants with very different characteristics: a relatively small onshore farm at a site with moderate terrain complexity, and a large offshore one in close proximity of the coastline. In both cases, the data-driven correction and tuning of the grey-box model results in much improved prediction capabilities. The identified flow fields reveal the presence of significant terrain-induced effects in the onshore case, and of large direction and ambient-condition dependent intra-plant effects in the offshore one. Analysis of the coordinate transformation and mode shapes generated by the singular value decomposition help explain relevant characteristics of the solution, as well as couplings among modeling parameters. CFD simulations are used for confirming the plausibility of the identified flow fields.

## 1 Introduction

Understanding and modeling wind farm flows is one of the key grand challenges facing wind energy science (Veers et al., 2019). The problem is extremely complex, because wind farm flows are driven by a number of interconnected physical phenomena, which are not only difficult to model, but also in part still poorly understood.

Within this very wide field, the present work tries to explore the idea of using the whole wind plant as a distributed sensor that, interacting with the atmospheric boundary layer, responds to it and, consequently, effectively measures the flow developing within its own boundaries. Exploiting this idea, can data from a wind plant be used to detect significant features in the flow, in support of an improved understanding of key driving phenomena? Can the same data be leveraged to derive more accurate flow models? And how can the knowledge already encapsulated in existing models be combined with the information contained in the data?



These questions are explored here in relation to engineering wake models.

## 1.1   Engineering wake models and their limitations

Within the plethora of wind farm flow models that have been developed, engineering wake models have carved an extremely successful niche for themselves at the lower end of the fidelity spectrum. In fact, they now support a wide range of use cases,
from wind plant design to wind farm flow control (Meyers et al., 2022). Engineering wake model suites, as for example FLORIS (NREL, 2021) or PyWake (Pedersen et al., 2019), are based on the bottom-up concept of superimposing relatively simple flow elements, such as wake deficit, wake added turbulence, wake deflection, wake combination, etc. The success of engineering wake models is due to their modularity, which allows for a rapid uptake of any new improvement of the individual sub-models, and to their speed, which is key in supporting repetitive tasks such as design, control, uncertainty quantification,
and others.

   However, as all models, engineering wake models are not an exact copy of reality, and are unable to precisely match field measurements. For example, the comprehensive survey of Lee and Fields (2021) showed that, although modeling techniques have greatly improved in recent years, inaccuracies in the estimation of the turbine inflow speed is still the largest contributor to the uncertainty in yield assessments.

A first reason for the mismatch between predictions and reality is the unsuitable calibration of the model parameters. For example, wake recovery is affected by atmospheric conditions and terrain roughness (Abkar et al., 2016; Wu and Porté-Agel, 2012), which depend on location and on time (for example, because of seasonal vegetation changes). Therefore, default standard values of the parameters describing recovery might not be appropriate for a specific site, nor for a specific time at that site.

Additionally, engineering wake models only approximate (but do not exactly resolve) only some (but not all) physical processes that take place in and around a wind pant.

   For example, the influence of terrain orography is difficult to capture for onshore wind farms, and high-fidelity models may be necessary to adequately resolve all flow effects (for example, see Berg et al. (2011) for the Bolund site, and Palma et al. (2020) for the Perdigao site). Neglecting terrain modeling can indeed increase the uncertainty of predictions generated by
engineering models (Fleming et al., 2019; Doekemeijer et al., 2021). FLORIS, which is the model used in this work, currently does not include a terrain flow solver, but offers the possibility of interpolating a set of wind speeds provided at different locations, for example obtained from met masts (Farrell et al., 2020). However, even with the inclusion of this heterogeneous background flow, the model still assumes wakes to follow the ground contour, and neglects the fact that terrain features may induce pressure gradients, local deviations of wake trajectories, changes in dissipation rate, flow separations, and other effects
(Porté-Agel et al., 2020; Politis et al., 2012; Castellani et al., 2017). Terrain and ground roughness can also affect offshore sites. For example, a doppler radar deployed at the Westermost Rough wind farm (Nygaard and Newcombe, 2018) and CFD simulations at Anholt (van der Laan et al., 2017) revealed the development of heterogeneous flow fields caused by the influence of the neighboring coastlines.



The interaction of a wind farm with the atmospheric boundary layer (ABL) is another extremely complex process, not yet always properly accounted for in engineering models. In general, several flow regions can be distinguished around and within a wind farm (Porté-Agel et al., 2020). Upstream, the induction zone is a region of decreasing flow speed, causing a phenomenon termed "blockage" (Wu and Porté-Agel, 2017; Segalini and Dahlberg, 2020), which has been observed through production data (Bleeg et al., 2018) and by long-range lidar measurements (Schneemann et al., 2021). Efforts at modeling this effect include the aggregation of individual turbine induction zone models (Nygaard et al., 2020; Branlard et al., 2020), and the use of the linearized Navier-Stokes equations (Segalini, 2021). Moving further downstream, an internal boundary layer starts growing over the turbines. If the farm is deep enough in the streamwise direction, the flow may reach a fully developed state, sometimes referred to as the "deep-array" condition (Calaf et al., 2010). In the fully developed region, momentum is only replenished by the vertical transport from the free atmosphere flowing above the farm. Theoretical models for an asymptotic flow regime have been developed under the assumption of an infinitely large wind farm (Frandsen et al., 2006). However, it is not clear at which distance from the leading edge a fully developed flow regime is reached (Wu and Porté-Agel, 2017); additionally, models typically assume a regular wind farm layout, a condition that is often not met in practice. The flow has complex features not only within the wind plant, but also at its perimeter. In fact, the flow has been reported to accelerate as it turns around the edges of a wind farm (Mitraszewski et al., 2013). Furthermore – similarly to hills, mountains and other large orographic features – wind farms can generate gravity waves, which not only affect the flow high above the turbines, but also cause a redistribution of the available wind resource within the plant. All of these phenomena appear to be strongly dependent on the ABL height and on atmospheric stability, stable conditions typically amplifying their effects (Wu and Porté-Agel, 2017; Allaerts and Meyers, 2018; Schneemann et al., 2021).

Yet another poorly understood and modeled effect is the way wakes interact, mix and merge together. Current models range from simple superposition laws, e.g. the sum of squares freestream superposition method (SOSFS) (Katic et al., 1986)) or the frestream linear superposition one (FLS) (Lissaman, 1979)), to more sophisticated physics-based combination models (Zong and Porté-Agel, 2020; Bastankhah et al., 2021), to methods that describe how wakes merge with the background flow (Lanzilao and Meyers, 2022). Despite these advances, new modeling proposals will take time for validation and adoption, while the simple SOSFS and FLS superposition laws are still heavily relied on. In conditions where many wake interactions take place, such models can have a substantial influence on the results (Hamilton et al., 2020).

## 1.2 Improvement of engineering models by data-driven tuning and learning

There are three main approaches to deal with the deficiencies of current engineering wake models.

The first is to eliminate the resolved part of the model altogether, and use a black box to learn the complete system behavior from data. Indeed, data-driven machine learning methods are a growing trend in many fields, including fluid mechanics (Brunton et al., 2020). Wind energy applications are no exception to this trend: for example, Göçmen and Giebel (2018) and Bleeg (2020) have proposed black-box farm flow models based on neural networks. While this approach seems appealing at first sight, it also neglects the large body of knowledge and experience already encapsulated in existing models. Additionally, trying to distill new understanding and physical insight from black boxes is in general not a trivial task. More importantly, one




should never forget that the data informational content always caps what a purely date-driven model can deliver: what is not in the data, can never be learnt. As a consequence, large amounts of data are typically necessary to derive useful models.

The second approach is to improve a (white) model by tuning its parameters. For example, van Beek et al. (2021) tuned the parameters of the FLORIS model using operational data, which resulted in a substantial error reduction. However, tuning the resolved physics in a model when relevant unresolved phenomena are present, may lead to nonphysical results. In this sense, one should be wary of approaches that only tune the parameters of an existing model, unless one can guarantee that there are no relevant missing physical effects in that model. As previously argued, this is typically not the case with present engineering

wake models.

The third possible approach it to directly acknowledge the hybrid nature of the problem. This means augmenting the resolved model with parametric corrections that represent the unmodeled physics, resulting in the so-called grey-box approach. Data is used to tune the parameters of the resolved model, and to learn the ones of the corrections. These two processes of tuning and learning are clearly intimately linked, and should be conducted simultaneously. In the framework of wind farm

flows, the approach of simultaneous tuning and learning (STL) was first proposed by Schreiber et al. (2019). The concept was demonstrated by augmenting the FLORIS wake model (NREL, 2021; Fleming et al., 2020) with various "surgical" ad hoc corrections, designed to represent non-uniform inflow, secondary steering and other unmodeled effects. The method has since been applied also to a wind tunnel study (Campagnolo et al., 2022), and to a joint flow model comparison (Göçmen et al., 2022). The resulting – possibly highly redundant – parameter estimation problem was solved using a maximum likelihood ap-

proach based on the singular value decomposition (SVD). The role of this decomposition is to map the original correlated and redundant physical parameters into uncorrelated ones. This simplifies the understanding of which parameters can be identified based on the informational content of the data, and which are undiscernible. Once the visible parameters are identified, they are transformed back into the physical ones through the inverse map.

In the present paper, the STL approach is extended by augmenting a wake model with an heterogeneous background flow,

which can be considered as a correction to the normally assumed uniform ambient flow. This way, the whole wind plant becomes a distributed sensor that "feels" the flow that develops within its boundaries; this has suggested the name of "farm as a sensor" to this approach. The similar concept of the "the turbine as a sensor" has been developed by the senior author and his collaborators, where a wind turbine is turned into a sensor that "feels" the inflow at its rotor disk; interested readers can refer to Bertelè et al. (2017); Schreiber et al. (2020); Bertelè et al. (2021) and references therein.

The paper is organized as follows. Section 2 describes the wind farm flow model (§2.1), its parameterization (§2.2), and the identification technique used to tune and learn the free model parameters from operational data (§2.3). Section 3 describes the application to an onshore wind farm at a site of moderate complexity (§3.1), and to a large offshore plant (§3.2). Finally, Sect. 4 reports the main findings of this work, and provides an outlook towards further future developments.



## 2 Methods

### 2.1 Wind farm flow

#### 2.1.1 Temporal decomposition

Within a wind plant, the scalar wind speed field $U$ at some reference height can be decomposed in the time domain as

$$U = \bar{U} + \tilde{U} + U'. \tag{1}$$

The term $\bar{U}$ represents a constant-in-time component. The term $\tilde{U}$ accounts for the slow temporal variability caused by changes in ambient conditions and in the turbine set-points, and their advection downstream throughout the plant. Finally, the term $U'$ accounts for fast fluctuations caused by turbulence.

Engineering models such as FLORIS (NREL, 2021) provide only for a steady-state (as opposed to time-resolved) representation of a turbulent wake immersed in a turbulent flow. Nonetheless, it is important to realize that the long-term effects of $U'$ are indeed included in such models. In fact, the wake geometry in the model is represented by an "average" path and shape, observed over a long-enough period of time. This way, the wake model implicitly includes the effects of meandering caused by turbulent fluctuations in the wind field. Additionally, the model accounts for the effects of both the local ambient and wake-added turbulence intensity (TI), noted $I = \mathrm{std}(U')/\bar{U}$, which affects the behavior (and especially the recovery) of the wake. The inclusion of these effects in the model also help clarify the split between the slow scales $\tilde{U}$ and the fast scales $U'$, and provides guidelines on where in the frequency spectrum one ends and the other one begins. In fact, a wake model is calibrated by fitting it to observations that have been averaged over a certain window of time (typically, equal to 10 minutes). Consequently, $\tilde{U}$ represents all the slower time scales that have not already been taken into account by this time averaging. Such scales are neglected in a steady-state model, and explicitly considered in a dynamic one (e.g., see the FLORIdyn dynamic wake model (Gebraad and van Wingerden, 2014)).

This work considers the steady-state behavior of wind plants for given ambient and operating conditions. Consequently, the wind field model includes only the component $\bar{U}$ (which, as just argued, implicitly includes the effects of the turbulent component $U'$), whereas $\tilde{U}$ is neglected. For notation simplicity, in the following the bar notation is dropped and the steady state wind field is simply noted $U$.

#### 2.1.2 Causal decomposition

The wind speed field can be causally decomposed as

$$U = U_{\mathrm{amb}} + \Delta U_{\mathrm{wake}} + \Delta U_{\mathrm{amb} \leftrightarrow \mathrm{wake}}. \tag{2}$$

The first term $U_{\mathrm{amb}}$ represents the undisturbed ambient flow at the site, in the absence of the wind turbines and their induced effects. This component of the flow depends on the state of the ABL and on the surface conditions, the latter including the effects of local orography (onshore) and of local roughness (caused by vegetation and small-scale terrain features in the on-



shore case, and by sea state in the offshore one). This distinction of surface causal effects can also be seen as a further scale
decomposition, the largest spatial scales being attributed to orography, and the smallest ones to roughness.

The second term $\Delta U_{\mathrm{wake}}$ represents the change in speed caused by wakes, as modeled by FLORIS or similar models. This
flow component depends on the state of the ABL, on the local ambient conditions at each turbine (including the possible
presence of wakes released by upstream machines), and on the turbine characteristics and their operating set-points.

The third and last term $\Delta U_{\mathrm{amb}\leftrightarrow\mathrm{wake}}$ represents the interaction between the undisturbed ambient flow and the one generated
by the turbines, and it can be further decomposed as

$$\Delta U_{\mathrm{amb}\leftrightarrow\mathrm{wake}} = \Delta U_{\mathrm{amb}\rightarrow\mathrm{wake}} + \Delta U_{\mathrm{wake}\rightarrow\mathrm{amb}}. \tag{3}$$

The term $\Delta U_{\mathrm{amb}\rightarrow\mathrm{wake}}$ accounts for the effects of the ambient background flow on the wake. It should be noted that sev-
eral of these effects are already included by design in engineering wake models: for example, ambient turbulence intensity
(Bastankhah and Porté-Agel, 2014), vertical shear and veer (Sezer-Uzol and Uzol, 2013) are known to affect the wake charac-
teristics, and their modeling approximations are included in FLORIS (NREL, 2021). Hence, all of these effects, and possible
future refinements designed to better reflect the influence of the characteristics of the ABL on wake behavior, already appear
in the term $\Delta U_{\mathrm{wake}}$. Therefore, the term $\Delta U_{\mathrm{amb}\rightarrow\mathrm{wake}}$ is tasked here with representing only the modifications to the wake
trajectory and shape caused by the heterogeneity of the ambient flow (Bossanyi, 2018), and by terrain orography and rough-
ness changes. These effects are neglected in the following, simply because – at the time of writing – the corresponding models
are not implemented in FLORIS; consequently, the term $\Delta U_{\mathrm{amb}\rightarrow\mathrm{wake}}$ is dropped from the discussion. However, when these
models finally become available, their presence will not alter the rest of the present formulation.

The term $\Delta U_{\mathrm{wake}\rightarrow\mathrm{amb}}$ represents the effects caused by the plant, i.e. the turbines and their wakes, on the ambient undis-
turbed flow. These include both intra-plant (array) effects (which, for example, cause the average flow to slow down within the
plant (Calaf et al., 2010), and to locally accelerate in between turbines (McTavish et al., 2015)), and extra-plant effects (which
cause the growth of a boundary layer over the plant, and result in blockage (Porté-Agel et al., 2020) and local edge effects
(Mitraszewski et al., 2012)).

In summary, the causal decomposition of the flow speed expressed by Eq. (2) can be re-written as

$$U = U_0 + \Delta U_{\mathrm{wake}} + \Delta U. \tag{4}$$

The first term, $U_0$, is the average uniform (i.e., spatially constant) wind speed. The second term $\Delta U_{\mathrm{wake}}$ represents the wake
deficit model, as implemented in FLORIS or similar tools. The third term is an heterogeneous correction that writes

$$\Delta U = \Delta U_{\mathrm{amb}} + \Delta U_{\mathrm{wake}\rightarrow\mathrm{amb}}, \tag{5}$$

where $\Delta U_{\mathrm{amb}}$ accounts for ambient surface-induced effects, and $\Delta U_{\mathrm{wake}\rightarrow\mathrm{amb}}$ for the effects of the turbines and wakes on the
ambient flow. In this paper, both of these causes are aggregated into one single correction term $\Delta U$. Disentangling these two
effects should in principle be possible by using suitable datasets, as only the latter depends on the turbine operating conditions.
A similar causal decomposition is assumed for wind direction $\Gamma$ and for turbulence intensity $I$. In fact, for both of these flow
characteristics similar arguments apply, as both can exhibit an heterogeneous behavior induced by surface effects and by the





interaction of the flow with the turbines and their wakes. Hence, noting a generic field as $F$ (where $F = U$, $F = \Gamma$, or $F = I$), the assumed causal model is written as

$$F = F_0 + \Delta F_{\mathrm{wake}} + \Delta F. \tag{6}$$

For given ambient and operating conditions, $F_0$ is a site-average (i.e. spatially constant) flow condition (either speed, direction or turbulence intensity). $\Delta F_{\mathrm{wake}}$ is the wake model; at present, in addition to the speed deficit, FLORIS includes secondary steering for $F = \Gamma$, and wake-added TI for $F = I$. Finally, $\Delta F$ is an heterogeneous (spatially variable) correction field. When considering the wind direction field, i.e. $F = \Gamma$, the correction needs to be applied in a circular manner with modulus $360°$.

The functional dependency of the heterogeneous correction term $\Delta F$ is assumed to be in the form

$\Delta F = \Delta F\left(\mathbf{A}_0, Q\right),$ (7)

where $\mathbf{A} = (U, \Gamma, I, L)^T$ is a vector of ambient state variables, $L$ is the Obukhov length, $(\cdot)_0$ indicates an average (spatially constant) quantity, and $Q$ indicates a spatial location. Hence, the correction term $\Delta F$ depends on:

– The site-average ambient conditions $\mathbf{A}_0$. In fact, different wind speeds, directions and turbulence intensities induce different interactions of the ambient flow with the surface and the plant.

– The spatial position $Q$, because surface conditions (including both orographic and roughness effects) and plant-induced phenomena are typically heterogeneous.

– The turbine set-points. However, it should be noted that this dependency is already implicitly taken into account by the dependency of $\Delta F$ on $\mathbf{A}_0$ and $Q$, because turbine set-points depend on local ambient conditions. An extra parameter could be used to account for different operating modes (for example, a quiet mode for nighttime operation), but it is 205 neglected for simplicity here, also because it is not used in the application examples.

## 2.2 Model parameterization

It is the primary goal of this paper to present a method for computing a best estimate of the flow fields expressed by Eq. (6), based on operational data. For given ambient conditions $F_0$, this requires first expressing the terms $\Delta F$ and $\Delta F_{\mathrm{wake}}$ in terms of free parameters (which is discussed in §2.2.1 and §2.2.2, respectively), and then estimating the values of such parameters 210 based on an optimality criterion, using available field measurements (which is explained in §2.3). In principle, the identification should ensure the satisfaction of fluid conservation properties for the resulting field expressed by Eq. (6); such constraints are however neglected in the present implementation. Once the values of the parameters have been computed, the resulting identified model can be used for performing new predictions in support of various use cases.

### 2.2.1 Heterogeneous flow parameterization

The spatial heterogeneity of field $\Delta F$ over the farm area is discretized using a 2D mesh, where the value of the field at a generic point $Q$ is obtained by interpolating discrete nodal values $\mathbf{p}_F$ through assumed shape functions $\mathbf{n}(Q)$. Notice that this



implies that the site is assumed to be flat, consistently with the current FLORIS model; as a result, the e.g. speedup caused by a hill is represented as a patch of increased velocity. The 2D spatial interpolation is expanded in additional 1D dimensions, to capture the influence of the environmental conditions $\mathbf{A}_0$. For example, when considering the effects of ambient wind direction

variability $(\Gamma_0)$, the range of wind directions is discretized into a desired number of nodal direction values, and assumed 1D shape functions are used to interpolate such values. This results in a different set of spatial speed nodes for each wind directional node, creating a 3D interpolation of flow speed accounting for spatial position and wind direction. This dependency of the interpolating functions on space and ambient conditions is expressed in symbols as $\mathbf{n}(\mathbf{A}_0, Q)$. Terrain and plant-induced effects can generate different heterogeneities in the speed, direction and turbulence intensity fields. Hence, different meshes

with different resolutions and node locations can in principle be used for each one of the three fields.

The parameterization of the $\Delta F$ field can be written as

$$\Delta F(\mathbf{A}_0, Q) = \mathbf{n}^T(\mathbf{A}_0, Q)\mathbf{p}_F . \tag{8}$$

The spatial dependency of $\Delta F$ is implemented in FLORIS through the heterogeneous flow functionality introduced in Farrell et al. (2020).

Alternatively, the heterogeneous field $\Delta F$ can also be defined as

$$\Delta F(\mathbf{A}_0, Q) = \Delta F_{\mathrm{NP}}(\mathbf{A}_0, Q) + \mathbf{n}^T(\mathbf{A}_0, Q)\mathbf{p}_F . \tag{9}$$

Here, the first term is a non-parametric (i.e., which will not be identified) heterogenous flow field. This term could be obtained from on-site measurements (Farrell et al., 2020) or, as shown later on in §3.1.6, from over-the-terrain CFD simulations. When this term is used, the parametric term $\mathbf{n}^T(\mathbf{A}_0, Q)\mathbf{p}_F$, instead of being charged with the modeling of the complete heterogeneity

of the flow, has the role of modeling only differences between the non-parametric flow field and the actual one. The inclusion of the non-parametric term might have two beneficial effects on the identification: first, it reduces the magnitude of the heterogeneity that has to be learnt from data, and, second, it provides a non-uniform initial guess for the identification algorithm, possibly easing its convergence.

### 2.2.2 Wake model parameterization

The $\Delta F_{\mathrm{wake}}$ component of the flow is computed through the FLORIS engineering wake model (NREL, 2021). The velocity deficit $\Delta U_{\mathrm{wake}}$ is modeled with the kinematic Gaussian model by Bastankhah and Porté-Agel (2014). Wakes are combined with the SOSFS method (Katic et al., 1986); the effects caused by different formulations of the wake combination sub-model are considered in §3.2 by comparing SOSFS with FLS (Lissaman, 1979). Wake-added turbulence $\Delta I_{\mathrm{wake}}$ is considered through the Crespo and Hernandez (1996) turbulence model.

In general, FLORIS and similar models are characterized by the following functional dependency

$$\Delta F_{\mathrm{wake}} = \Delta F_{\mathrm{wake}}(\mathbf{A}_0, Q, \mathbf{k}) , \tag{10}$$

where $\mathbf{k}$ represents a vector of model-specific parameters (NREL, 2021). Following Schreiber et al. (2019), the model-specific parameters are not tuned directly; rather, the baseline value (noted $k_{\mathrm{init}}$) of one parameter is added to an unknown calibration



term (noted $p_k$), i.e.

$$k = k_{\mathrm{init}} + p_k \ . \tag{11}$$

All calibration parameters $p_k$ are collected in the vector of to-be-tuned parameters $\mathbf{p}_W$. Examples of the parameters and their changes caused by calibration are given later on (see Table 2 in §3.1.3, and Table 4 in §3.2.6).

Notice that, in addition to the "native" parameters of the FLORIS model, additional extra parameters can be used to augment the model with ad-hoc correction terms. Schreiber et al. (2019) used this technique to target specific deficiencies of the model.

For example, the baseline wake model was augmented with a local wind direction term to account for secondary steering, which was not natively implemented at the time in FLORIS. Similarly, Campagnolo et al. (2022) introduced a correction to the power loss model for yawed conditions.

## 2.3  SVD-supported identification

Stacking the parameters for the heterogeneous flow correction and the parameters for wake model tuning, the final vector of

to-be-identified parameters writes

$$\mathbf{p} = \begin{bmatrix} \mathbf{p}_U \\ \mathbf{p}_\Gamma \\ \mathbf{p}_I \\ \mathbf{p}_W \end{bmatrix} . \tag{12}$$

Notice that one single parameter vector is defined, comprising both the parameters that define the unknown heterogenous flow and the one that tune the wake model. This means that learning of the heterogeneous flow is performed simultaneously to wake model tuning. In fact, if one were to estimate the two components $\Delta F$ and $\Delta F_{\mathrm{wake}}$ one after the other, any error committed

in the estimation of the first would affect the second, and the results would be sequence dependent. Since this cannot be, given that both terms eventually contribute to the fields $F$, the two terms need to be estimated simultaneously.

Following a classical approach (Jategaonkar, 2015), a likelihood function is used to express the probability that a given set of noisy observations be explained by a specific set of parameters. The parameter identification problem is then cast as the maximization of this likelihood function. This problem, however, is very likely ill posed. First, it is uncertain if all parameters

are really observable given the existing measurements. Additionally, the parameters might not all be independent of each other, resulting in similar effects on the solution.

This dilemma is overcome by performing the identification through a singular value decomposition (SVD). The SVD-supported identification approach is general, and can be applied to various problems: for example, Bottasso et al. (2014) used it for identifying airfoil polars, and Schreiber et al. (2019) for learning unrepresented effects in a wind farm flow model.

The main idea behind this method is to map the original physical parameters into uncorrelated ones, using a linear rotational transformation of the problem unknowns that is made possible by the SVD. Examination of the new set of parameters reveals the ones that are identifiable – because they have an acceptably low variance –, and the ones that are not – because their





variance is excessively large. Only the former parameters are retained in the process and, once they have been identified, they are mapped back through the inverse transformation, recovering a solution in terms of the original physical parameters. Since

many practical identification problems are non-linear, this linear transformation of the unknowns is applied iteratively, until convergence. This method has the advantage of working well even in the presence of unobservable or collinear parameters, simply because only the visible ones are retained in the process. Additionally, inspection of the transformation that maps the original into the uncorrelated parameters reveals useful insight on the interdependencies among parameters; an example of such an analysis is given later on in §3.1.4.

For a complete derivation of the method, the reader is referred to Schreiber et al. (2019), whereas here only a synthetic description is provided.

A steady wind farm flow model can be written as the nonlinear functional expression

$$\mathbf{y} = \mathbf{f}(\mathbf{p}, \mathbf{A}) \,, \tag{13}$$

where $\mathbf{y}$ indicates a vector of model outputs for which corresponding measurements $\mathbf{z}$ are available. In the present work, these

quantities are represented by the power outputs of $N_t$ wind turbines in a wind plant; however, the definition of the outputs is clearly problem dependent, so that other measurements – when available – could be readily used. The un-inevitable discrepancy between measurements $\mathbf{z}$ and model outputs $\mathbf{y}$ is captured by the residual, which is defined as

$$\mathbf{r} = \mathbf{z} - \mathbf{y} \,. \tag{14}$$

Given a set of $N$ observations $\{\mathbf{z}_1, \mathbf{z}_2, ..., \mathbf{z}_N\}$, the likelihood function (Jategaonkar, 2015) writes

$$J(\mathbf{p}) = \left( (2\pi)^{N_t} \det(\mathbf{R}^{-1}) \right)^{-N/2} \exp\left( \frac{1}{2} \sum_{i=1}^{N} \mathbf{r}_i{}^T \mathbf{R} \mathbf{r}_i \right) \,, \tag{15}$$

where $\mathbf{R}$ is the measurement noise covariance matrix. The maximum likelihood estimate (MLE) of the parameters is obtained by minimizing the negative logarithm of Eq. (15), i.e.

$$\mathbf{p}_{\mathrm{MLE}} = \arg\min_{\mathbf{p}} -\ln J(\mathbf{p}) \,. \tag{16}$$

The observability of the parameters can be gauged by the inverse of the Fisher information matrix $\mathbf{E} \in \mathbb{R}^{N_p \times N_p}$, which is

defined as (Jategaonkar, 2015)

$$\mathbf{E}^{-1} = \left[ \sum_{i=1}^{N} w_i \left[ \frac{\partial \mathbf{y}_i}{\partial \mathbf{p}} \right]^T \mathbf{R}^{-1} \left[ \frac{\partial \mathbf{y}_i}{\partial \mathbf{p}} \right] \right]^{-1} = \mathbf{P} \,. \tag{17}$$

Factors $w_i$ express an optional relative weight $w_i / \sum_{j=1}^{N} w_j$ that can be attributed to an observation, to boost its presence in the dataset, for example because it recurs multiple times (Karampatziakis and Langford, 2010). The $i$-th diagonal element of $\mathbf{P}$ provides a lower bound (called Cramér-Rao bound) on the variance of the corresponding estimated parameter, while

correlations among different parameters are captured by the off-diagonal terms. This is indeed the main problem of a naive





formulation of the identification problem cast in term of the original physical parameters $\mathbf{p}$: even if a parameter has a high variance, typically it can not be eliminated because of its (in general) non-negligible couplings to other parameters. This problem is eliminated when the SVD is used to diagonalize the inverse Fisher matrix $\mathbf{P}$ by a linear transformation of the unknowns. Since the transformed parameters are now uncorrelated, parameters that have a high variance – i.e., that are not

visible given the available set of measurements – can be readily eliminated. Diagonalization of the inverse Fisher matrix is obtained by first factorizing it as $\mathbf{E} = \mathbf{M}^T\mathbf{M}$, where factor $\mathbf{M} \in \mathbb{R}^{N_t N \times N_p}$ writes

$$\mathbf{M} = \begin{bmatrix} \sqrt{w_1}\,\mathbf{R}^{-1/2}\frac{\partial \mathbf{y}_1}{\partial \mathbf{p}} \\ \sqrt{w_2}\,\mathbf{R}^{-1/2}\frac{\partial \mathbf{y}_2}{\partial \mathbf{p}} \\ \vdots \\ \sqrt{w_N}\,\mathbf{R}^{-1/2}\frac{\partial \mathbf{y}_N}{\partial \mathbf{p}} \end{bmatrix} . \tag{18}$$

This matrix is now decomposed by the SVD as the product

$$\mathbf{M} = \mathbf{U}\boldsymbol{\Sigma}\mathbf{V}^T , \tag{19}$$

where $\mathbf{U} \in \mathbb{R}^{N_t N \times N_t N}$ and $\mathbf{V} \in \mathbb{R}^{N_p \times N_p}$ are the left and right singular vectors, respectively. The matrix of left singular vectors $\mathbf{U}$ expresses the relative importance of the individual observations, while the matrix of right singular vectors $\mathbf{V}$ carries information on the correlation of the parameters. Matrix $\boldsymbol{\Sigma} = [\mathbf{S}, \mathbf{0}]^T$ contains the singular values $s_i$, which are sorted in descending order in the diagonal matrix $\mathbf{S} \in \mathbb{R}^{N_t \times N_t}$. By combining Eq. (19) and the factorization of $\mathbf{E}$, the eigendecomposition of the inverse Fisher matrix can now be written as

$$\mathbf{P} = \mathbf{V}\mathbf{S}^{-2}\mathbf{V}^T . \tag{20}$$

This suggests a transformation of the original parameters $\mathbf{p}$, which are rotated through matrix $\mathbf{V}$ to yield a new set of parameters $\boldsymbol{\theta}$, i.e.

$$\boldsymbol{\theta} = \mathbf{V}^T\mathbf{p} . \tag{21}$$

Crucially, the covariance of the new parameters is now $\mathbf{S}^{-2}$, which by definition is a diagonal matrix. Consequently, the new

parameters are statistically decoupled. Their respective variance $s_i^{-2}$ is readily obtained by the corresponding element in $\mathbf{S}$.

Next, an observability threshold $\sigma_t^2$ is introduced to define the highest acceptable variance, a condition that is enforced as $s_i^{-2} \leq \sigma_t^2$. This leads to a partitioning into identifiable (noted with the subscript ID) and non-identifiable (noted with the subscript NID) parameters, i.e. $\boldsymbol{\theta} = \left[\boldsymbol{\theta}_{ID}{}^T, \boldsymbol{\theta}_{NID}{}^T\right]^T$, which induces a corresponding partitioning of the transformation matrix $\mathbf{V} = [\mathbf{V}_{ID}, \mathbf{V}_{NID}]$ (and, clearly, also of $\mathbf{U}$). The MLE identification is then performed for the sole identifiable parameters

$\boldsymbol{\theta}_{ID}$. At the end of the process, the orthogonal parameters are mapped back to the physical ones using

$$\mathbf{p} \approx \mathbf{V}_{ID}\boldsymbol{\theta}_{ID} . \tag{22}$$



For guiding the solution, it is useful to enforce bounds on the parameters in the form $\mathbf{p}_{lb} \le \mathbf{p} \le \mathbf{p}_{ub}$. Additionally, to improve conditioning, it is advisable to scale each parameter $p_i$ with its respective bounds as

$$\hat{p}_i = \frac{p_i - (p_{i_{ub}} + p_{i_{lb}})/2}{(p_{i_{ub}} - p_{i_{lb}})/2} \,, \tag{23}$$

so that $-1 \le \hat{p}_i \le 1$.

## 3   Results

The result section is divided in two parts, each examining a specific site. The Sedini and Anholt wind farms represent a typical mid-size onshore and large offshore case, respectively. These two plants are characterized by different wind climates and dominating flow effects, whose very distinct features are useful for assessing the generality of the proposed STL method.

Furthermore, the quality and quantity of SCADA data typically differ from site to site, on account of different turbine types, acquisition systems, sampling frequencies, failure rates, miscalibration of sensors, and several other effects; here again, the use of different plants can help verify the robustness of a method that operates based on operational data of such variable quality and quantity. An overview of some key characteristics of the two wind plants is provided in Table 1, while Fig. 1 shows their layouts side by side, illustrating their typical spacings and overall size.

**Table 1.** Comparison of the main characteristics of the Sedini and Anholt wind farms.

| Name | Turbine model | Installed units | Rated power [MW] | Diameter [m] | Hub height [m] | Farm size N-S×W-E [km] | Typical spacing [D] |
|---|---|---|---|---|---|---|---|
| Sedini | GE1.5s/GE1.5sle | 36/7 | 1.5/1.5 | 70.5/77 | 65/80 | 3×2 | 2.3–9 |
| Anholt | SWT-3.6-120 | 111 | 3.6 | 120 | 81.6 | 22×12 | 5–10 |

The Sedini wind farm is located in the north of Sardinia, a large island off the western coast of Italy. A subgroup of turbines was the subject of a wind farm flow control test campaign, using both wake steering and axial induction control. Because of this previous activity, the behavior of the farm had been already examined with different wake models (Bossanyi and Ruisi, 2020; Doekemeijer et al., 2021). At this site, the terrain is complex, both outside and within the boundaries of the wind plant, vegetation is present, and the turbines are of two different types and heights, all characteristics that make of the Sedini wind

plant a challenging onshore test case. The farm is designed for minimum wake losses in the prevalent (westerly) wind direction, and significant wake effects are only expected for specific turbines. In addition to the layout, Fig. 1a shows also the grid of flow correction nodes, which was based, for simplicity, on a regular mesh. Node spacing was adjusted to capture the most relevant terrain effects.

      The Anholt offshore wind farm is located about 20 km east of the Danish coast in the Kattegat, a shallow sea between the

Jutland peninsula and the west cost of Sweden. The presence of the Jutland coastline influences the western inflow to the farm, creating a gradient that was already investigated by van der Laan et al. (2017), Peña et al. (2017) and Doekemeijer et al. (2022). Given the absence of the small-scale orographic effects present at Sedini, a coarser grid of flow correction nodes was chosen





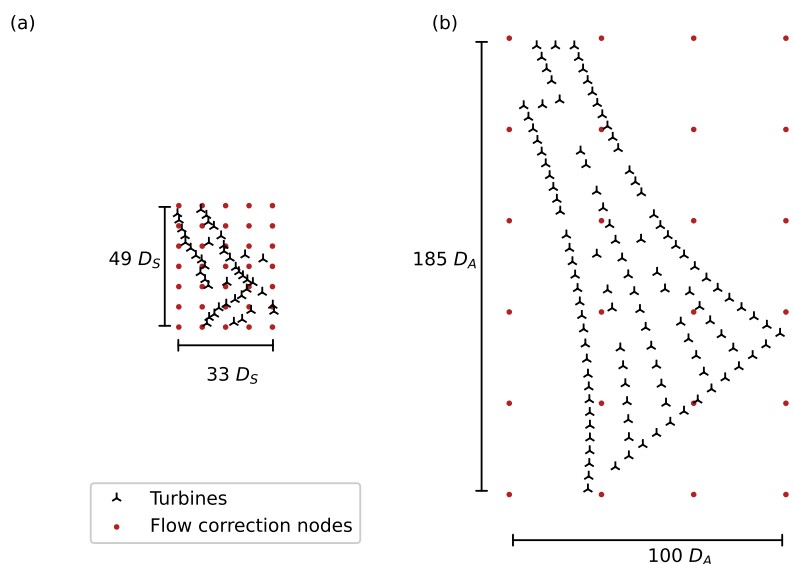

**Figure 1.** Layout and flow correction grid for the Sedini **(a)** and Anholt **(b)** wind farms. The symbol $D_S$ stands for the rotor diameter at Sedini, and $D_A$ for the diameter at Anholt, whose respective values are given in Table 1. The two figures are at the same scale in terms of diameters, i.e. 1 $D_S$ on the left panel has the same length as 1 $D_A$ on the right one. At the same kilometer scale, the Sedini farm looks much smaller than the Anholt one, because $D_A \gg D_S$.

in this case. On the other hand, this large array with numerous wake interactions is an interesting test case for the presence of significant intra and extra-plant effects (Nygaard, 2014).

### 3.1 The Sedini wind farm

#### 3.1.1 Site overview

Figure 2 shows a more detailed layout of the plant, including the turbine identifiers and a colormap of the terrain elevation. The behavior of the plant is investigated for the main, western, wind sector. From this direction, the farm is only two rows deep and the heterogeneous correction of Eq. (5) is expected to be dominated by the term $\Delta F_{\mathrm{amb}}$. The goal of the present test case 365 is therefore to show the ability of the proposed method of learning the orography-induced inhomogeneities of the intra-plant flow purely from the available SCADA data.

For this study, SCADA data at 10-minute sampling frequency was made available for the years 2015 and 2016, whereas meteorological mast measurements were made available for the years 2008–2010. Since the two time periods do not overlap, the mast data was used only to analyse the general climate at the site (Kern et al., 2017).

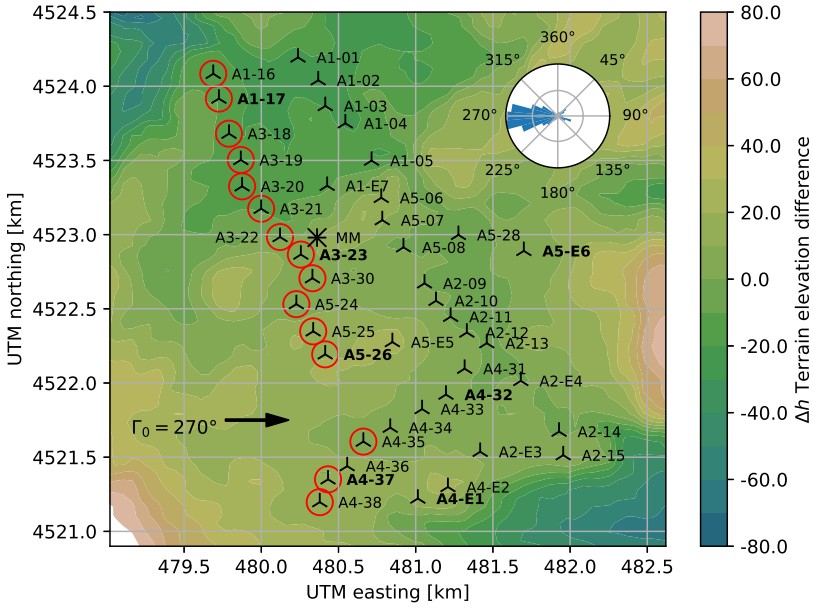

**Figure 2.** Layout of the Sedini wind farm. The colormap shows the height difference with respect to the average terrain elevation. A bold identifier indicates turbines used to determine the average wind direction $\Gamma_0$. For an exemplary wind direction of $270°$, free-stream turbines are marked with a red circle. The wind rose shows the frequency of $\Gamma_0$ over the period of time analyzed in the present study. The met mast is indicated by the symbol $*$.

The data was first cleaned of entries where turbines were not reporting to the acquisition system. Next, for every timestamp, the average wind direction $\Gamma_0$ was determined from the yaw readings of selected "sensing" turbines. This required a careful correction of the readings, for yaw sensors were observed to be significantly affected by biases and drifts. These effects were mitigated by exploiting wake interactions among turbines. In fact, biases were eliminated by looking at the minima of the power ratio between neighboring waked/waking turbines as functions of wind direction, and comparing them with the interactional

directions expected from the farm layout. Drifts were eliminated by observing the shift over time of these minima, and removing them from the time series. Notwithstanding these corrections, since the yaw readings of some turbines appeared to be quite unreliable, only a cluster of eight machines was finally used to determine the wind direction.

Because short-term fluctuations $\tilde{F}$ are neglected in this work, data preprocessing was performed to obtain binned observations dominated by the steady component $F$, according to the methods described by Schreiber et al. (2018). To omit short-term

propagation effects, a stationary filter was applied to the data streams. Similarly to Hansen et al. (2012), a data point was discarded if the wind direction change exceeded $\pm 2.5°$ compared to the previous 10-minute value. The ambient wind speed $U_0 = \langle U_{FS} \rangle$ was determined by averaging the rotor equivalent wind speed (REWS) of turbines operating in free stream. The determination of whether a turbine is in free stream or not was based on the prediction of the plant flow model, by first guessing the wind speed and then iterating. The seven turbines of type GE1.5sle (whose identifiers in Fig. 2 contain the letter "E") were





not used for determining the wind speed; since they are characterized by a taller hub height than the other ones, this would have required the use of the vertical shear, introducing further uncertainties.

Additional ambient conditions such as TI, shear and density – although certainly significant for wake behavior and turbine performance and loading – cannot be typically derived in a straightforward manner from the sole turbine SCADA data. Göçmen and Giebel (2016) and Mittelmeier et al. (2017) proposed methods to deduce TI and density from SCADA data, but

unfortunately the necessary channels were not available in the present case. Although met mast measurements were obtained over a different time period of time than the operational SCADA data, they were used to try to address this limitation of the dataset. In fact, inspection of the met mast recordings suggested that TI and shear are strongly dominated by the diurnal cycle at this site. Based on this indication, the dataset was split in daytime and nighttime regimes, based on the local time of sunset and sunrise (Beauducel, 2022). The diurnal characteristic values were derived from historical met mast readings. For daytime

conditions the shear was set to $\alpha_0 = 0.09$ and the TI to $I_0 = 0.15$, whereas for the nighttime case the values $\alpha_0 = 0.18$ and $I_0 = 0.125$ were used. Density was set to the constant average value $\rho_0 = 1.177$ kgm$^{-3}$. After filtering, the remaining 2102 timestamps were grouped and averaged in day/night bins of width equal to 5° for wind direction and 2 ms$^{-1}$ for wind speed. This resulted in $N = 36$ observations for the wind direction sector 245°–310°. Half of the bins were picked randomly to form the training dataset, whilst the other half was reserved for testing.

**3.1.2  STL parameter identification**

The output vector **y** (cf. Eq. (13)) was defined as the normalized generated power of every turbine, i.e.

$$\mathbf{y} = \frac{1}{P_{\text{ref}}} \begin{bmatrix} P_{WT1} \\ \vdots \\ P_{WT43} \end{bmatrix} , \tag{24}$$

where $P_{\text{ref}} = 1.5$ MW is the rated power of the GE turbines. Power was calculated in 1°-wide direction steps, eventually averaging the results over each 5° bin sector.

The STL parameter vector **p** was defined as follows:

– For the data-driven learning of the heterogeneous background velocity $\Delta U = \Delta U_{\text{STL}}$, a north-oriented, regular mesh of 5×7 flow correction nodes was superimposed onto the farm (see Fig. 1a). The node spacing is 470 m and 450 m in the eastern and northern directions, respectively. It was verified that a finer spatial discretization did not improve the quality of the results. An additional discretization of the environmental conditions $\mathbf{A}_0$ (see Eq. (7)) was performed

only for wind direction. To this end, a second set of nodes was placed every 15°, i.e. for the distinct values $\Gamma_0 \in [255°, 270°, 285°, 300°]$. Results indicated that the relative correction $\Delta U/U_0$ does not change significantly depending on wind speed. This suggests that the terrain flow is Reynolds independent, as often assumed in over-the-terrain CFD applications (van der Laan et al., 2020). Therefore, each nodal correction parameter $p_{U,i}$ was treated as a non-dimensional speedup factor, independently of the inflow wind speed. To accommodate this change, Eq. (4) was re-written as

$$U = U_0 \left( 1 + \Delta \hat{U} \right) + \Delta U_{\text{wake}} , \tag{25}$$





where $\Delta\hat{U} = \Delta U/U_0$ is now a relative correction. The term $I_0$ was also found to have no significant influence on the results, and was therefore omitted from the dependencies of the flow speedup. According to these choices, the heterogeneous background velocity was discretized using 140 unknown nodal values $\mathbf{p}_U$. The relative speedup bounds were set to $\pm 0.3$, i.e. the corrections can change the reference speed by $\pm 30\%$.

– Although orography-induced effects may in principle result in the heterogeneity of the wind direction at a site, such an effect could not be observed at Sedini based on the available dataset. On the other hand, a global correction of the wind direction proved to be necessary and very beneficial for the quality of the results. This was achieved by using a single correction node $p_\Gamma$ in Eq. (8), without any assumed dependency on $\mathbf{A}_0$. This resulted in a shift of the wind direction, constant throughout the entire farm area and independent of the ambient conditions, which was learnt from

the operational data of the turbines. It was not possible to clarify with certainty the root reason for this offset, which is probably due to some problem with the yaw sensors.

    – The identification of a heterogeneous TI field $\Delta I$ was omitted, because the available SCADA data did only contain 10-minute power min/max values (sometimes used as a proxy for TI (Mittelmeier et al., 2017)), nor other information that could be used for this purpose.

– The wake model behavior is captured by the wake parameter vector $\mathbf{p}_W$, which includes the four velocity parameters $\alpha$, $\beta$, $k_a$, and $k_b$, and the four turbulence parameters $I_{\text{constant}}$, $I_{\text{ai}}$, $I_{\text{initial}}$, and $I_{\text{downstream}}$. The wake model parameters were tuned within the range $\pm k_{\text{init}}$, simultaneously to the learning of the flow correction parameters $\mathbf{p}_U$ and $p_\Gamma$.

These choices led to the definition of the unknown parameter vector $\mathbf{p} = [\mathbf{p}_U, p_\Gamma, \mathbf{p}_W]^T$, resulting in a total of $N_p = 149$ to-be-identified parameters.

The error covariance matrix was assumed to be known a priori and diagonal, i.e. $R_{i,j} = \sigma_m^2 \delta_{i,j}$, where $\delta_{i,j}$ is the Kronecker delta. This assumption can be eliminated by estimating the covariance from the residuals, and iterating until convergence (Jategaonkar, 2015), although this did not significantly improve the results in the present case. The cost function expressed by Eq. (15) was selected as

$$J(\mathbf{p}) = \frac{1}{2} \sum_{i=1}^{N} w_i \mathbf{r}_i{}^T \mathbf{R} \mathbf{r}_i \, , \tag{26}$$

where the factors $w_i$ are proportional to the number of 10-minute observations within each bin, in order to weight their participation based on the number of samples that they contain. The observability threshold for the orthogonal parameters was set to $\sigma_t^2 = 0.1^2$, which corresponds to a 2% variance in the scaled range $[-1, 1]$.

     The solution procedure was based on first applying the SVD, thereby recasting the STL-parameters into the orthogonal set $\boldsymbol{\theta}$. After discarding the orthogonal parameters whose variance fell above the observability threshold, the optimization was

run with the Sequential Least SQuares Programming (SLSQP) minimization algorithm, as implemented in the SciPy library (Virtanen et al., 2020). This process was repeated three times to ensure convergence, as expressed by changes in the singular values $s_i$. Figure 3 shows the distribution of the variance of the orthogonal parameters, i.e. the squared inverse of the singular



values, before the third and last run of the MLE algorithm. Of the 149 orthogonal parameters, 94 had a variance below the threshold and were retained in the optimization; the number of retained parameters was constant throughout the iterations. The same figure shows that 24 parameters exhibit extremely high variances. These parameters are associated to flow correction nodes that lie outside of the perimeter of the farm; since the identification process is purely driven by data that is co-located with the turbines, the parameters associated with these farm-external nodes carry very little information, and hence have very high variance. The informational content of the retained singular values can be estimated as $\phi = \sum_{i=1}^{N_r} s_i^2 / \sum_{i=1}^{N_p} s_i^2 = 97\%$, where $N_r = 94$ is the number of retained parameters (Golub and van Loan, 2013).

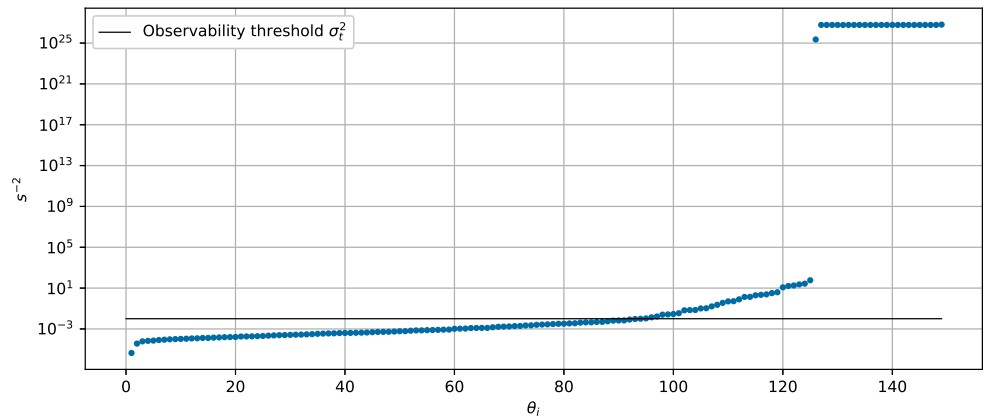

**Figure 3.** Variance of the orthogonal parameters before the last MLE. Only 94 parameters were retained in the identification, whereas the others above the variance threshold were discarded.

### 3.1.3 Results for wind direction correction and wake model tuning

As previously mentioned, the wind direction was corrected over the entire domain by the value $\Delta\Gamma = 5.6°$, suggesting that the average yaw sensor readings are affected by an offset. The exact reason for such a large difference could not be ascertained and might be due to a combination of factors, including the small number of turbines with acceptable yaw signals (just eight), possible miscalibrations of the sensors, and the manually-performed correction of drift and biases based on wake interactions. In this sense, the ability of the STL approach of automatically finding the optimal correction appears to be very useful, since sensor biases – especially in the yaw drives– are a common challenge (Bromm et al., 2018).

Table 2 reports the results of the wake model tuning. According to Eq. (11), the initial baseline values $k_{\text{init}}$ are summed to their respective correction parameters $p_k$ to yield the final, tuned model parameters $k$. The extent of the near wake region is determined by $\alpha$ and $\beta$, while $k_a$ and $k_b$ model the wake expansion in the far region (Bastankhah and Porté-Agel, 2016). Examining the relative parameter changes, reported in the last row of the table, it appears that $\alpha$ – which models the influence of turbulence intensity on the downstream extension of the near wake – is the term that changed the most. With the present tuning, for an ambient $I_0 = 0.14$, the near wake is 28% shorter than with the initial baseline values. With an earlier start of



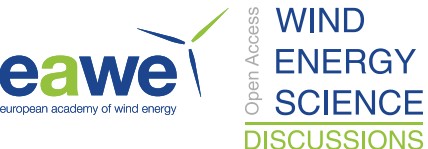

decay, the far wake deficit is reduced. As a result, a downstream turbine operating at about 3 D (the typical distance for the Sedini farm) produces 42% more power, thereby significantly decreasing the wake losses predicted by the baseline tuning. Furthermore, the wakes have a higher sensitivity to the different day/night ambient turbulence intensity than before. On the other hand, tuning led to an only marginal increase of the added turbulence model.

**Table 2.** Results of the wake model tuning, with the initial baseline parameters $k_{\text{init}}$, the identified values of the additive corrections $p_k$ and the final tuned parameters $k$. The last row reports the relative change from $k_{\text{init}}$ to $k$.

| | $\alpha$ | $\beta$ | $k_a$ | $k_b$ | $I_{\text{constant}}$ | $I_{\text{ai}}$ | $I_{\text{initial}}$ | $I_{\text{downstream}}$ |
|---|---|---|---|---|---|---|---|---|
| $k_{\text{init}}$ | 0.58 | 0.077 | 0.38 | 0.004 | 0.9 | 0.75 | 0.5 | -0.325 |
| $p_k$ | 0.28 | 0.010 | 0.0 | 0.0 | 0.0 | 0.03 | 0.0 | -0.015 |
| $k$ | 0.86 | 0.087 | 0.38 | 0.004 | 0.9 | 0.78 | 0.5 | -0.34 |
| $\pm$ | 48% | 13% | - | - | - | 4% | - | -5% |

### 3.1.4 Orthogonal decomposition

An examination of the rotation matrix $\mathbf{V}$ can give some useful insight on the relative importance and correlation of the physical parameters. For a simpler visualization, the large $\mathbf{V} \in \mathbb{R}^{N_p \times N_p}$ matrix was condensed in a way that tries to capture the effects of the various types of corrections. According to Eq. (12), the overall parameter vector $\mathbf{p}$ is obtained by stacking the different correction parameter vectors, which induces an identical block-row partitioning of $\mathbf{V}$, i.e.

$$\mathbf{V} = \begin{bmatrix} \mathbf{V}_U \\ \mathbf{V}_\Gamma \\ \mathbf{V}_W \end{bmatrix} = \begin{bmatrix} \mathbf{V}_{U255°} \\ \vdots \\ \mathbf{V}_{U300°} \\ \mathbf{V}_\Gamma \\ \mathbf{V}_W \end{bmatrix} . \tag{27}$$

In the third term of the previous expression, $\mathbf{V}_U$ has been further partitioned by the directional bins. By definition, each singular vector, i.e. each column $\mathbf{v}_i$ of matrix $\mathbf{V}$, has unit length, i.e. $|\mathbf{v}_i| = 1$. A visual representation of the matrix that captures the overall contribution of each parameter type was obtained by taking the root sum of squares of each row block partition. The resulting reduced matrix is visualized in Fig. 4, where the columns are sorted in descending order of the associated singular values. The vertical dashed line represents the cutoff at 94 retained parameters (generated by the observability threshold), which divides $\mathbf{V}_{\text{ID}}$ and $\mathbf{V}_{\text{NID}}$, i.e. the identifiable from the non-identifiable orthogonal parameters.

Inspection of the reduced matrix suggests a few observations. First, the directional correction $p_\Gamma$, which applies a constant offset throughout the farm, is almost exclusively contained in the first singular vector. Since this bias affects every turbine over the entire dataset, it has a prominent position in the decomposition and it is essentially uncoupled from the other parameters. The wake model tuning parameters appear immediately behind the wind direction in the ranking, and they are also highly uncoupled from the rest. An examination of the individual rows of this block (not visible in the figure) shows that the largest

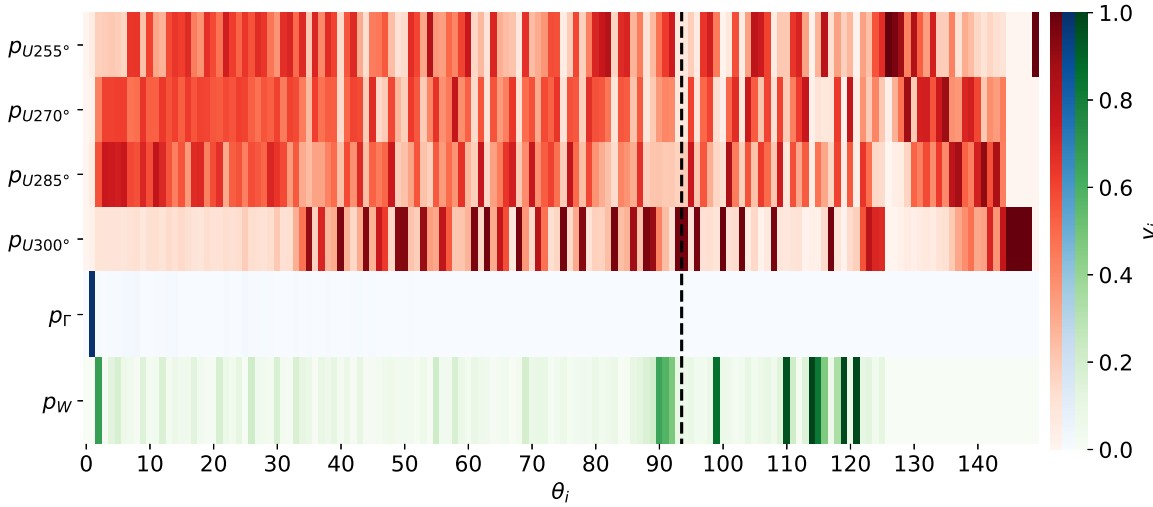

**Figure 4.** Reduced matrix $\mathbf{V}$ of singular vectors, obtained by taking the root sum of squares of the rows of each row block partition (corresponding to each different parameter type). The different colors represent the different correction categories: wind speed correction (red), wind direction correction (blue), wake model tuning (green).

contributions come from the $k_a$ and $\alpha$ parameters, as already shown by Table 2. The observability of these parameters improved by introducing the day/night variability of $I_0$, as previously explained. Finally, the appearance of velocity corrections in the singular vectors seems related not only to the heterogeneity of the flow, but also to the number of available observations. In fact, the largest number of data points is available around $270°$ and $285°$, whereas data is more sparse around $300°$, which results in flow correction parameters with lower observability.

To better understand the nature of the corrections $\Delta U$, the singular vectors can be mapped to the farm domain via their shape functions. This is obtained by combining Eq. (8) with Eq. (22), to yield

$$\Delta U = \mathbf{n}^T \mathbf{p}_U \approx \mathbf{n}^T \mathbf{V}_{U,ID} \boldsymbol{\theta}_{ID} = \boldsymbol{\Psi}_U^T \boldsymbol{\theta}_{ID} \; , \tag{28}$$

where $\boldsymbol{\Psi}$ is the matrix of eigenshapes (Bottasso et al., 2014). To facilitate the visualization of the eigenshapes, the discussion is restricted to the small sector $285° \pm 10°$. Figure 5 shows the relative decrease of the cost function (evaluated only in the subsector $285° \pm 10°$), when activating one orthogonal parameter at a time. The first reduction (blue shade in the figure) can be, as already stated, attributed almost exclusively to the directional correction $p_\Gamma$. Likewise, the second orthogonal parameter contains mostly corrections to the wake model (green shade). The parameters appearing after the wake model corrections are associated with flow speed corrections. For this subsector, parameters $\theta_{4-6}$ and $\theta_{11}$ are the most effective ones in reducing the cost function. They account for ca. 41% of the final cost function improvement and, therefore, are responsible for removing the largest heterogeneous flow discrepancies. Inspection of Fig. 4 shows that, for the row corresponding to $\mathbf{p}_{U,285°}$, the indices of these parameters are indeed associated with a large contribution to the matrix of singular vectors. The cost function is



essentially flat for the orthogonal parameters associated to directions outside of the subsector. For example, this is clearly the case for $\theta_{7-10}$ (corresponding to $\mathbf{p}_{U,255°}$).

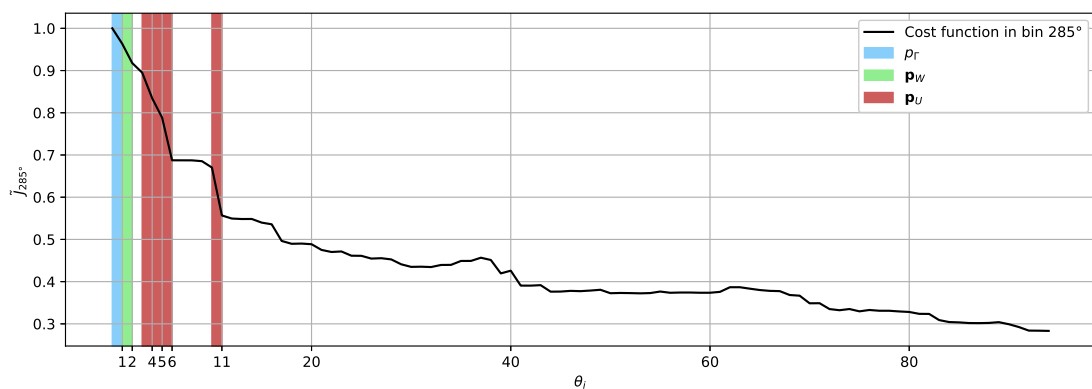

**Figure 5.** Decrease of the normalized subsector cost function when activating one orthogonal parameter at a time. The red-shaded areas represent the flow speedup corrections that contribute the most to the error reduction. The corresponding eigenshapes are visualized in Fig. 6.

Figure 6 finally shows the red-shaded eigenshapes $i = 4 - 6$, 11, superimposed onto the farm map. In addition to these dominating modes, the figure also reports the lowest flow-related eigenshape (corresponding to $i = 3$), although its cost function improvement is only modest (see Fig. 5). Each eigenshape is multiplied by the sign of the corresponding orthogonal parameter, i.e.

$$\boldsymbol{\psi}_i = \mathbf{v}_i^T \mathbf{n} \operatorname{sign}(\theta_i), \tag{29}$$

to show speedup and slowdown corrections in a consistent manner.

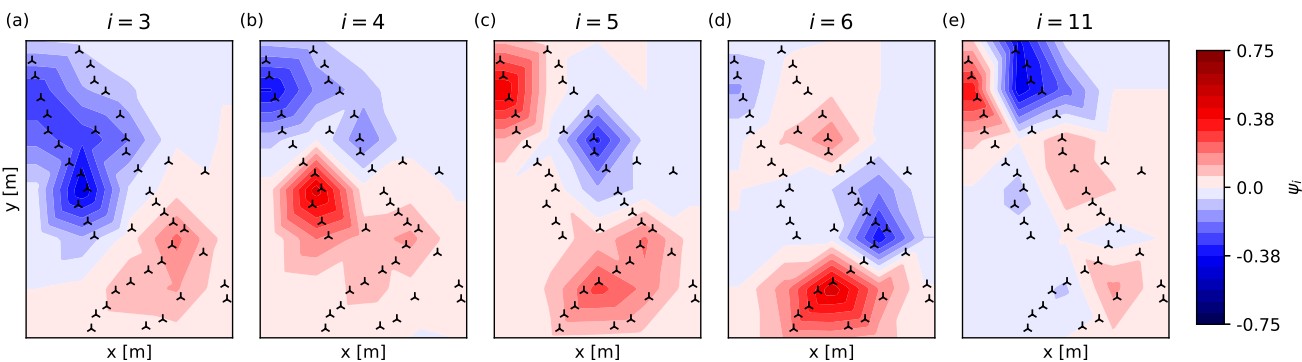

**Figure 6.** Dominating eigenshapes of the flow corrections $\Delta U$ in the $285° \pm 10°$ sector. With increasing counter $i$ (i.e. singular value), corrections become more fragmented, i.e. spatially localized.





The first eigenshape, Fig. 6a, represents a roughly north-south speed change. Comparing this plot to the terrain map of Fig. 2, shows that the ground elevation in the northern part of the farm is lower than in the south. As elevated regions generally induce higher velocities, this lowest mode captures this prominent orographic effect of the site. The higher-order eigenshapes become increasingly fragmented and seem to model specific localized terrain features. For example, $\psi_4$ (see Fig. 6b) captures the very prominent hill in the middle of the western row.

### 3.1.5 Plausibility check via CFD

The corrections identified by the proposed method describe a direction-dependent heterogeneous flow field that very significantly improves the matching of the FLORIS model predictions with measured operational data. However, is this identified flow field a reasonable approximation of the true flow over the terrain at this site, or is it just a mathematical correction that happens to improve the results? A definitive answer to this question is probably difficult to give with the limited data and information available. However, a qualitative verification of the plausibility of the field that was learnt from data can be obtained by comparing it with an independent CFD simulation of the flow over the terrain.

To perform this plausibility check, Reynolds-averaged Navier-Stokes (RANS) simulations were conducted for the values $\Gamma_0 \in [255°, 270°, 285°, 300°]$, without the turbines and in neutral atmospheric conditions. The resulting flow fields represent direction-dependent steady-state heterogeneous flows over the terrain, which can be directly compared with the learnt data-driven corrections. In principle, the latter also contain intra-plant effects induced by the turbines, which are not present in the former; however, as mentioned in §3.1.1, given the small streamwise extent of the Sedini farm for this sector, $\Delta U_{\text{wake}\rightarrow\text{amb}}$ effects are probably very small and hence negligible. Clearly, since there is no independent comparison of the CFD results with the actual flow, only a qualitative comparison between numerical and learnt corrections is possible. A more complete description of the setup of the RANS simulations is provided in Appendix A.

As previously stated, the flow is assumed to be Reynolds-independent. As for the STL-identified flow, learnt corrections are expressed in the form of non-dimensional speedup factors. The absolute velocity field was extracted from the computed flow field at hub height (65 m), and then normalized by the average speed at the free-stream turbine locations.

For the directions $255°$, $270°$, $285°$, and $300°$, Fig. 7 shows the learnt (left) and CFD-computed (right) speedup fields. Looking at the figure, it appears that there is a general agreement on the location of low and high speed regions. A comparison with the terrain map of Fig. 2 shows that elevated and depressed regions of the terrain are coherent with flow speed slowdowns and speedups. This suggests that the terrain elevation is the main driver of the identified corrections. The STL method seems to estimate more pronounced flow inhomogeneities than CFD. For example, this can be seen for the hill at the southern end of the farm, and for the lower speed region in the northern end. Additionally, the flow pattern in the northern region for direction $285°$ (Fig. 7e) corresponds to the correction introduced in eigenshape $i = 11$ in Fig. 6e. The CFD simulations further show some distinct speedup regions in the center of the farm, which are caused by small hills. For example, for the $255°$ direction, the STL method is probably not resolving all flow details correctly. This is a result of the coarse resolution of the grid and, more importantly, of the distance among turbines – and, thus, of the distance among learn-driving measurements – in that area. Furthermore, the extrapolation of the STL results outside of the farm perimeter cannot work well beyond a short distance,





because of a lack of information in the data that drives the learning process. Notice however that, from the point of view of the quality of the FLORIS model predictions, the lack of knowledge of the flow outside of the farm is of no importance, as long as
the inflow on the upstream turbines is correctly captured.

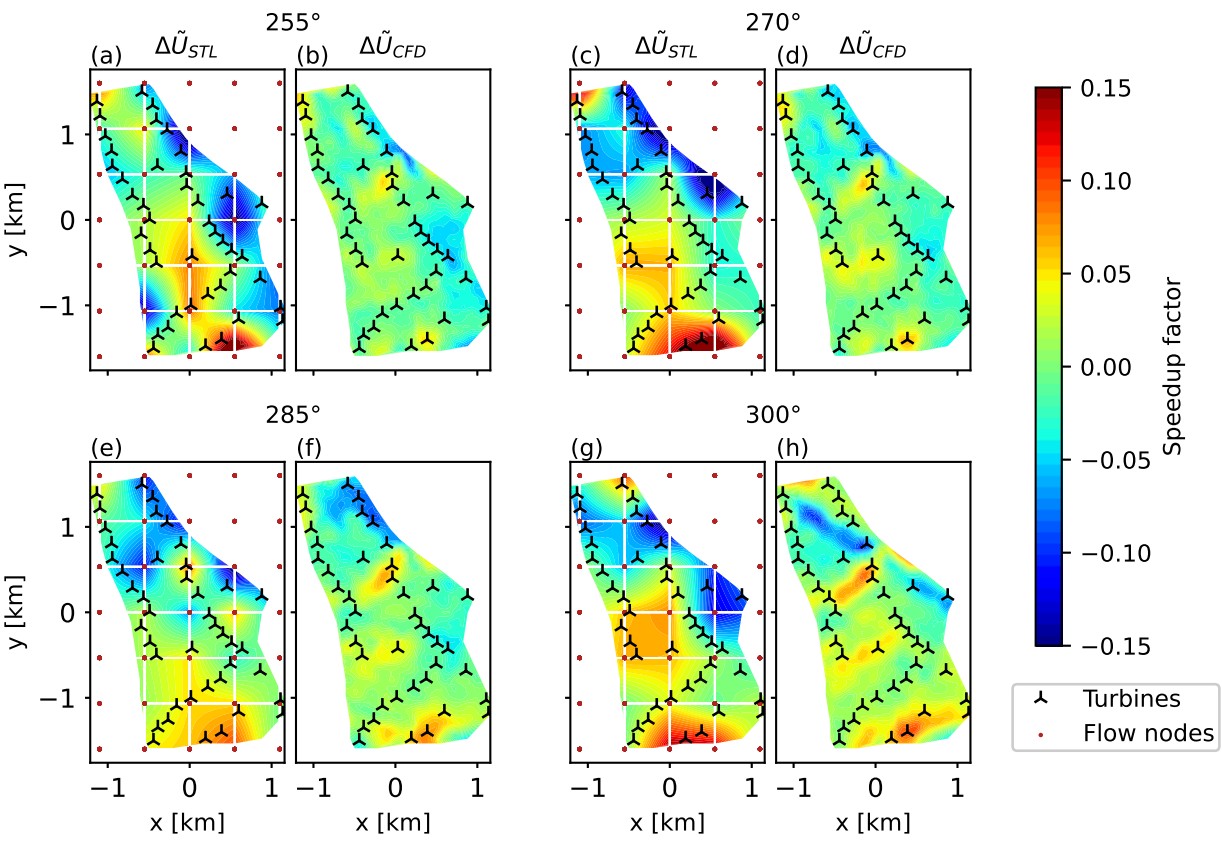

**Figure 7.** Comparison between the learnt (**a**, **c**, **e**, **f**) and CFD-computed (**b**, **d**, **f**, **h**) speedup factors, for directions $255°$, $270°$, $285°$, and $300°$. The learnt results are corrected by the identified directional offset $\Delta\Gamma$.

### 3.1.6   Initialization of STL by a CFD-computed field

Corrections can be learnt with respect to an initial heterogenous flow field, instead of a uniform one (i.e., utilizing Eq. (9) instead of Eq. (8)). To verify whether these better initial conditions can lead to improved results, the RANS CFD simulations were used to initialize the background flow, thereby providing a non-uniform baseline solution; in this case, the role of the
data-driven corrections is that of compensating any remaining discrepancies. The CFD-computed speedup factors for a generic wind direction were obtained by linear interpolation between the two adjacent simulated directions. The same definition of the flow correction nodes employed in the previous case was used here too.



Results indicate that the identified wind direction $p_\Gamma$ and wake $\mathbf{p}_W$ parameters (see §3.1.3) were not affected by the improved initial background flow. Figure 8 shows, for the $270°$ direction case, the CFD initial baseline solution (Fig. 8a), the learnt

corrections (Fig. 8b), and the final heterogeneous velocity field (Fig. 8c). The learnt corrections seem in general to increase the CFD-computed terrain inhomogeneities.

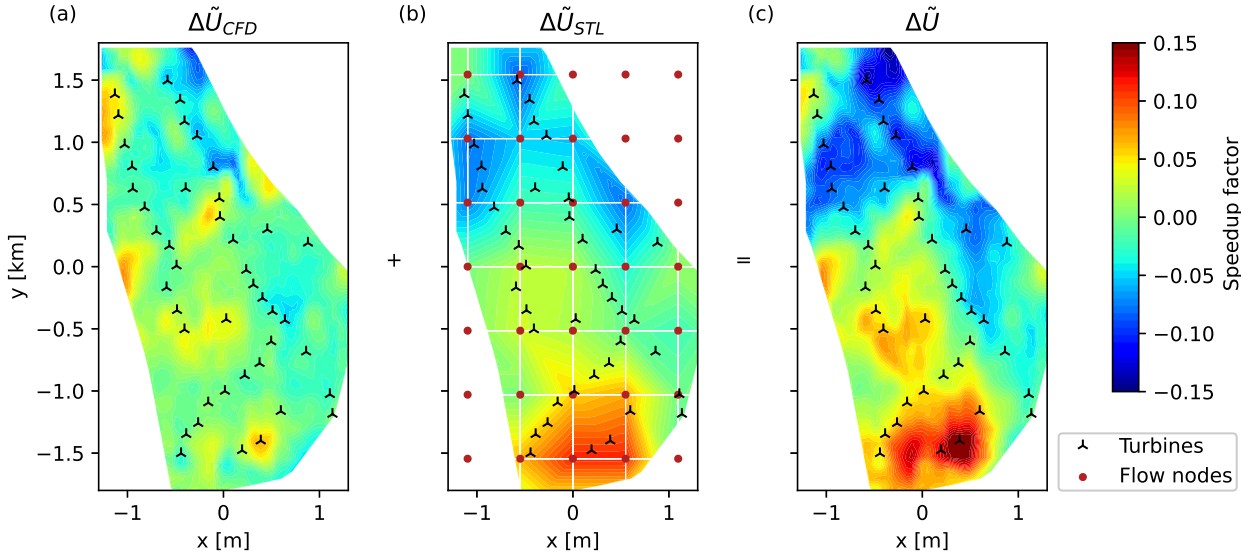

**Figure 8.** Initial CFD-computed heterogeneous flow (**a**); data-driven learnt correction field (**b**); final resulting flow field. All results are for the $270°$ wind direction.

As shown in the next section, the use of a CFD-computed initial flow field offers quantitatively no visible error reduction for power, when compared to the simpler option of starting from an initial uniform background flow. Indeed, the solution shown in Fig. 8c is very similar to the one of Fig. 7c, while the direction and wake correction parameters are also essentially identical.

As a consequence, the turbine inflows, where the error is computed, are very similar. However, the CFD-based approach allows for a finer resolution of the flowfield in between the turbines and externally to the farm perimeter. In addition, the fact that essentially the same solution is obtained for very different initial conditions, seems to indicate the absence of distinct local minima, at least for this case.

### 3.1.7 Contributors to the error improvement

At convergence of the estimation process, the remaining error is defined as

$$\epsilon = \frac{P_{\text{SCADA}} - P_{\text{model}}}{P_{\text{rated}}},\tag{30}$$

where $P$ is the overall farm power, and the error is calculated only using the test part of the dataset, i.e. discarding the data points used for learning. Four cases of increasing complexity were compared, as listed in Table 3. In the baseline case, the





FLORIS model is used without any correction, i.e. using a homogeneous background flow, no wind direction correction, and
wake-describing parameters sourced from the literature. In the case labelled "CFD", the background flow is the one computed
with the RANS model, without additional data-driven corrections; in this case, learning is limited to the wind direction and the
tuning of wake-describing parameters. In the case labelled "STL", the initial background flow is uniform, and learning is used to
compute the full heterogeneity of the flow, in addition to direction and wake behavior. Finally, in the case labelled "CFD+STL",
the initial background flow is the RANS-computed one, and learning is used to further correct this already heterogeneous field,
in addition to direction and wake behavior.

**Table 3.** Four different cases for the analysis of learnt corrections on the power matching error.

|  | $\Delta U_{\mathrm{CFD}}$ | $\Delta U_{\mathrm{STL}}$ | $\Delta \Gamma$ | Wake model tuning |
|---|---|---|---|---|
| Baseline | - | - | - | - |
| CFD | ✓ | - | ✓ | ✓ |
| STL | - | ✓ | ✓ | ✓ |
| CFD + STL | ✓ | ✓ | ✓ | ✓ |

Figure 9 gives an overview of the error reduction that can be achieved compared to the baseline performance. Figure 9a
shows the impact of each different correction type on the overall error $\epsilon_{\mathrm{rms}} = \sqrt{1/N_{\mathrm{test}} \sum \epsilon_i^2}$. For all considered cases, the
addition of an heterogeneous velocity field resulted in the most substantial error improvement. As expected, the error for the
"CFD" case is larger than in the cases when the background flow is learnt or corrected; in fact, since the CFD results are not
aware of any on-site measurements, they are probably not completely accurate and representative of the actual terrain-induced
inhomogeneities. The final error for the "STL" and "CFD+STL" cases is extremely similar, showing that – notwithstanding
the different initial conditions – the solution is essentially the same. The identified directional offset was similar in all three
cases, and thus decreased the error in the same manner. Given the strong effects caused at this site by the terrain-induced flow
heterogeneity and the significant direction bias, the wake model tuning accounted only for a relatively small improvement of
the error.

Figures 9b through 9d show the probability distribution of the errors for three binned wind speed regimes. The FLORIS
baseline model tends to overpredict power production for low wind speeds, and to underpredict it for high wind speeds. Using
STL, this effect is eliminated, and the error spread is significantly improved. As already noticed, "STL" and "CFD + STL"
achieve very similar error distributions.

Figures 10 and 11 give a more detailed insight in the learnt corrections. The figures show the $5°$ binned measurements and
calculated power per turbine in the sector $245°$–$310°$ during daytime operation, for the wind speed range $U_0 \in [6,8]\ \mathrm{ms}^{-1}$.

Figure 10 focuses on two turbines that clearly highlight the improvements achieved by learning. In particular, turbine A2-12
(Fig. 10a) experiences a distinct wake shading by turbine A5-E5 in the direction range 255-270° (cf. the farm layout shown in
Fig. 2). The corrected model matches well the behavior of the measurements, whereas the baseline model is visibly offset on
account of the large wind direction bias.

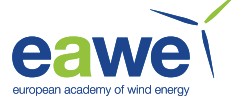



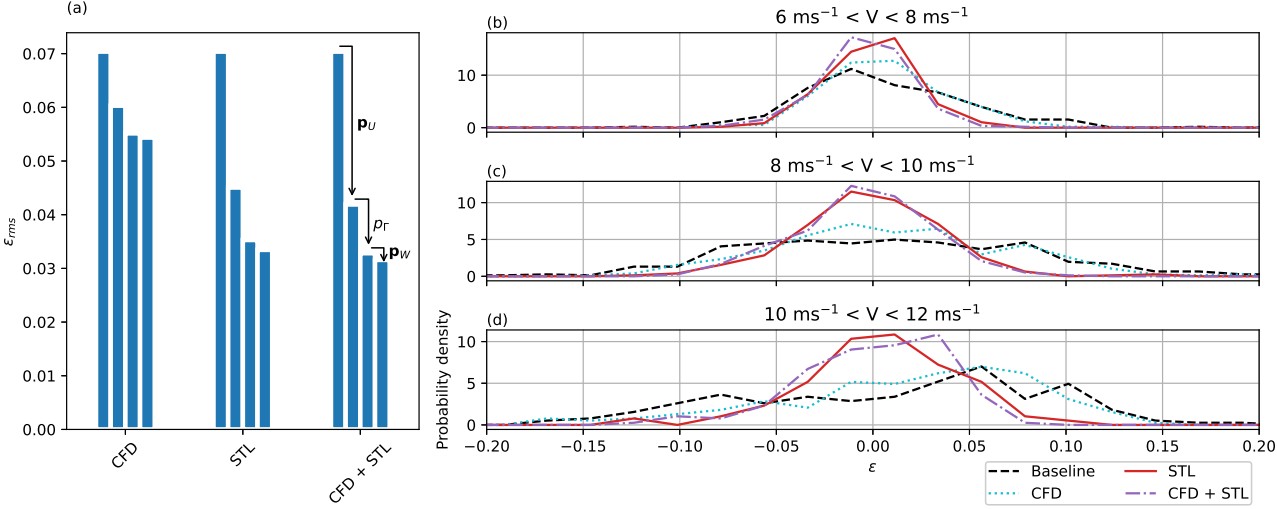

**Figure 9.** Reduction of the overall error by the activation of different correction types (**a**). Error probability density distribution for different wind speed ranges (**b–d**).

A similar situation is observed for turbine A1-E7 in the general overview plot of Fig. 11. Note that the power drops observed for $\Gamma_0 = 300°$ do not originate from wake interactions, but are due to a low bin average speed $U_0$. In fact, not many distinct wake interactions are visible in the plot, as the farm layout was specifically designed for this main wind direction. Some turbines in the western row are even operating in free stream conditions over the entire dataset; these machines are labelled FS

in Fig. 11. The effects of the terrain-induced flow corrections can be clearly appreciated by looking at turbine A5-E5 (Fig. 10b), which is located on a hill that is about 20 m higher than the farm average elevation. Without the heterogeneous flow model, FLORIS underestimates the power output of this turbine. However, in the "STL" case, the hill-induced speedup is captured by the learnt corrections. The speedup is also visible at the location of turbine A5-E5 in Fig. 7.

The color of the frames of each subplot of Fig. 11 shows the elevation difference $\Delta h$ of the turbine foundations with

respect to the farm average. The power of the turbines at the lower elevations is mostly overestimated by the baseline FLORIS, whereas the opposite happens for the turbines at the higher elevations. The learnt corrections compensate for these terrain-induced effects, leading to a good overall match throughout the whole plant. For completeness, Appendix B reports the results for nighttime operation.

## 3.2   The Anholt wind farm

### 3.2.1   Site overview

Figure 12b shows the Anholt wind farm together with its surrounding coastlines. The farm consists of 111 Siemens Gamesa SWT 3.6-120 wind turbines, and it is situated about 20 km east of the Jutland peninsula and about 25 km west of the island of





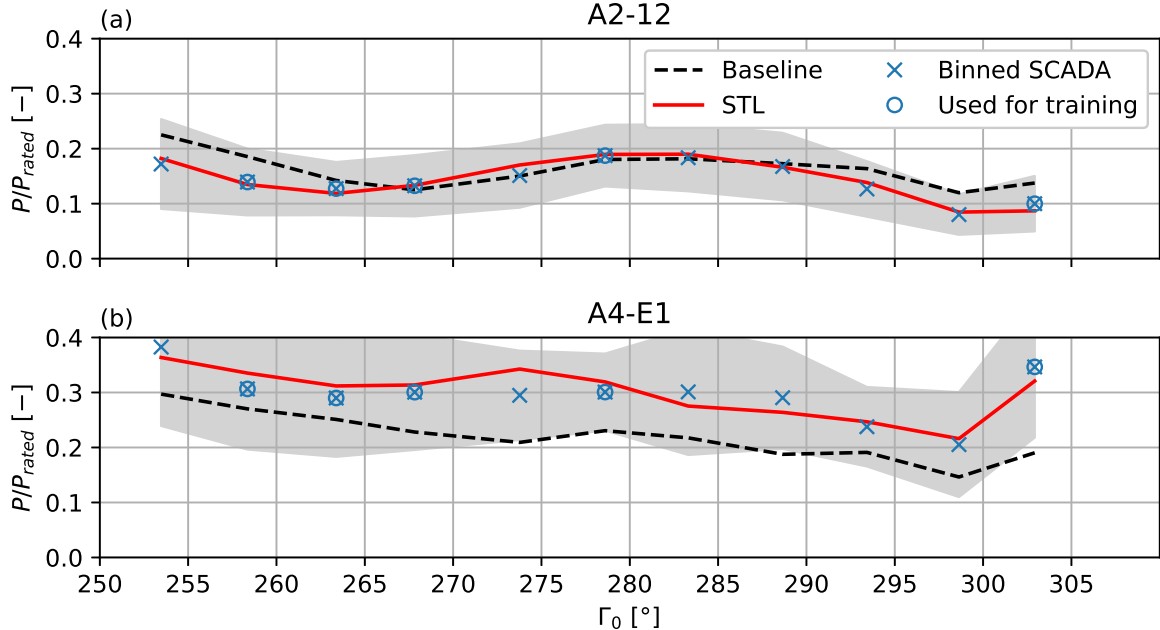

**Figure 10.** Normalized measured and calculated power for the two turbines A2-12 (**a**) and A4-E1 (**b**), for all $5°$ bins in the investigated western sector, for wind speeds in the range 6-8 ms$^{-1}$, during daytime operation. Bins with $< 10$ observations are not shown. Bins used for training are marked with a $\circ$ symbol. The uncertainty band shows the standard deviation in the bins. The baseline results are calculated without any data-driven corrections.

Anholt. The prevailing wind direction at the site is west/southwest. The farm has an irregular spacing, varying between 5 and 12 rotor diameters: the turbines forming the farm perimeter have a close spacing of about 5-6 D, whereas the spacing within the farm is larger.

Tuning and learning was performed using the same procedures as in the Sedini case. However, the two cases are significantly different, impacting the relative importance of the heterogeneous corrections terms of Eq. (5):

- Term $\Delta U_{\mathrm{amb}}$. The fact that Anholt is an offshore farm does not mean that terrain effects are absent. On the contrary, a terrain and roughness-induced velocity variation exists, caused by the land upstream of the site (whereas the effects of changing sea state were not considered). Hence, a heterogeneous velocity field can be identified from the turbine operational data. Similarly to the onshore case, even here a terrain-only CFD simulation provides a qualitative solution for verifying the plausibility of the data-driven corrections.

- Term $\Delta U_{\mathrm{wake \to amb}}$. The much larger streamwise depth of the farm increases the importance of plant-induced effects compared to the Sedini case. Such effects are expected to depend on the stability of the atmosphere (Porté-Agel et al., 2020), which here was approximately taken into account by binning based on mesoscale re-analyses.

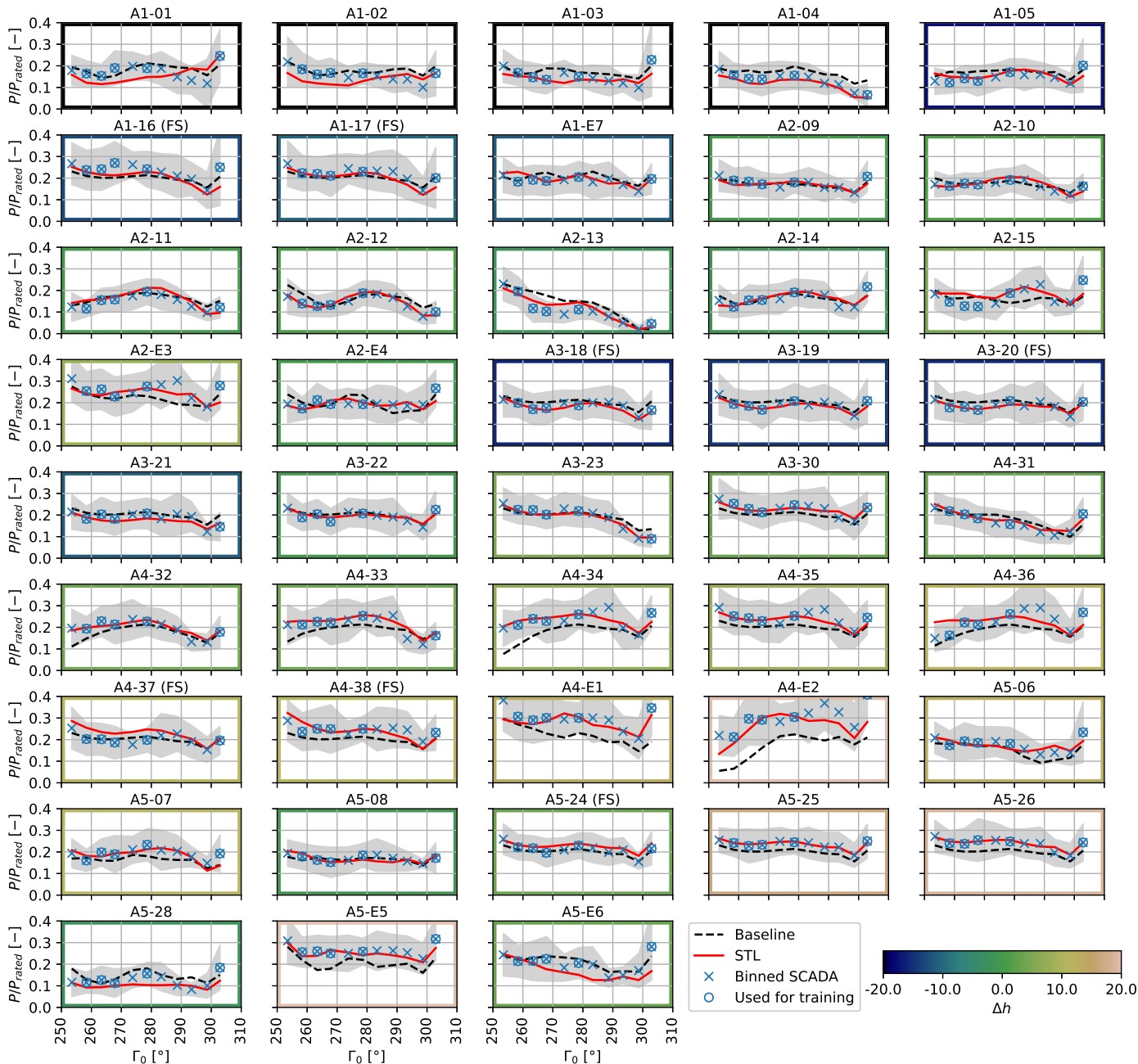

**Figure 11.** Normalized measured and calculated power for all turbines, for all $5°$ bins in the investigated $245°$-$310°$ sector, for wind speeds in the range 6-8 ms$^{-1}$, during daytime operation. Bins with $< 10$ observations are not shown. The uncertainty band indicates the standard deviation in the bins. $\Delta h$ is the foundation elevation difference with respect to the farm average, and is shown by the color of the frame of each subplot. Results during nighttime operation are given in Appendix B.





Since in this case both correction terms are relevant, it is not a straightforward task to disentangle the learnt plant-induced corrections $\Delta U_{\text{wake}\rightarrow\text{amb}}$ from the ambient ones $\Delta U_{\text{amb}}$. Although more sophisticated approaches are certainly possible, here a simple solution was adopted that consists in comparing the non-uniform inflow caused by the coastline with CFD analysis of the site. For brevity, the analysis of the SVD decomposition performed for Sedini is omitted in this case.

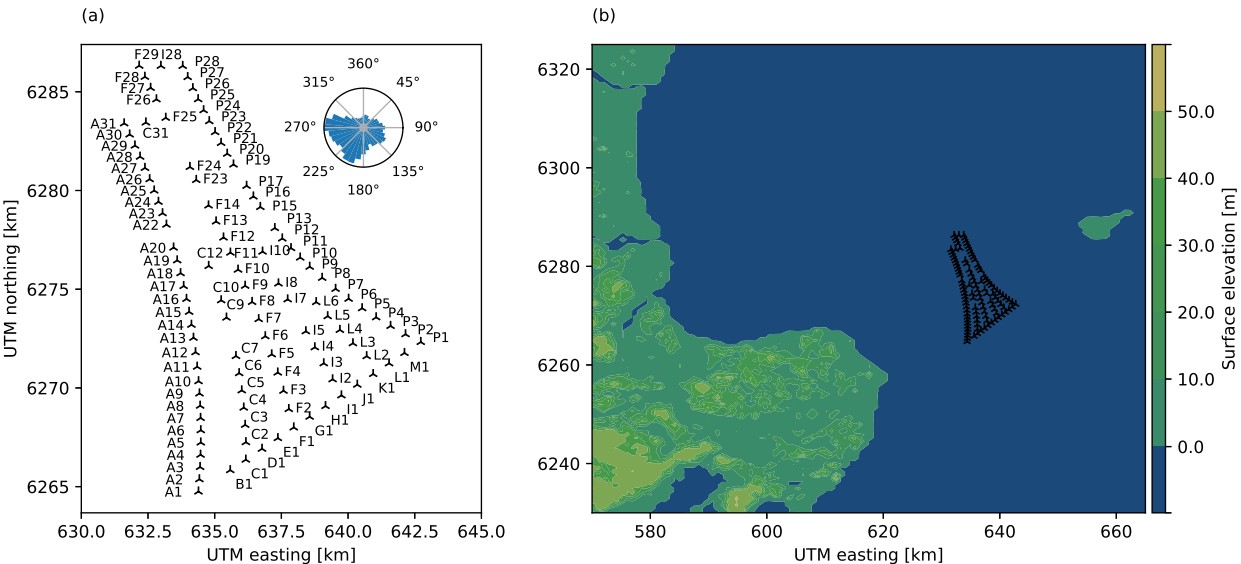

**Figure 12.** Layout of the Anholt wind farm with turbine identifiers and wind direction frequency (**a**). Location of the site, including the Jutland peninsula to the west and the island of Anholt to the east (**b**).

### 3.2.2 Data setup

For the present analysis, SCADA data at 10-minute sampling frequency was available from January 2013 until July 2015. The overall problem setup, solution methods and data preprocessing were the same used for the onshore plant, as described in §3.1. In contrast to the Sedini case, however, the data of the yaw sensors was found to be of a higher quality and consistency. Consequently, after removing yaw jumps and offsets, the ambient wind direction $\Gamma_0$ was calculated as the mean of the yaw signals across all turbines. The reference speed was determined by averaging the REWS of the free-stream turbines. Furthermore, a constant air density of $\rho_0 = 1.225$ kgm$^{-3}$ and a vertical wind shear exponent $\alpha_0 = 0.11$ were considered. The wind shear was derived as the average measured at the lidar buoy for unwaked directions. Note that, since all turbines have the same hub height, the shear has only a very modest effect on the results.

Mesoscale wind climate simulations of the region were carried out by Peña et al. (2017) using the Weather Research and Forecasting model (WRF, Skamarock et al. (2008)). The results are in the form of time series of hourly outputs for the years 2013-2015, corresponding to the SCADA time period, with an horizontal resolution of 2×2 km, interpolated at the turbine hub



height (81.5 m). The simulations do not include the effects caused by the wind turbines on the boundary layer (Fitch et al., 2012) .

In addition to providing a comparison for learnt coastline-induced effects, the WRF time series were used to filter the
dataset for atmospheric stability. Following Van Wijk et al. (1990), periods with Obukhov lengths in the range $0 < L \leq 1000$ were classified as stable, whereas periods with $-1000 \leq L < 0$ were labelled as unstable; neutral conditions were defined for $|L| > 1000$. Based on these criteria, 22% of the time stamps were classified as stable, and 64% as unstable; neutral conditions occurred only 7% of the times, and were therefore not considered further. This dominance of unstable conditions was observed at other Baltic offshore farms, e.g. Rødsand (Motta et al., 2005; Archer et al., 2016).

Unstable and stable observations were separated, creating two distinct datasets. Since turbulence intensity could be not inferred from the available SCADA data, it was assigned based on stability, using $I = 7.5\%$ for unstable and $I = 5\%$ for stable conditions; these values are based on met mast measurements at the Horns Rev wind farm, as reported by Hansen et al. (2012). Notice that the diurnal cycle, as utilized at the Sedini site, is not very dominant in offshore conditions (Motta et al., 2005). After filtering, the first dataset consisted of $17\,492$ unstable 10-minute time stamps, and the second of $3351$ stable ones. Both
datasets were grouped and averaged in direction and speed bins, respectively of $10°$ and $2 \text{ ms}^{-1}$ of width, resulting in $N = 108$ observations for both sets. Half of the bins were picked at random to form the training dataset, whilst the other half was reserved for testing.

### 3.2.3   STL parameter identification

The STL parameter vector $\mathbf{p}$ was defined as follows:

– Similarly to the Sedini case, the wind speed corrections were defined as Reynolds-independent speedup factors $\Delta\hat{U}$, as in Eq. (25), which were discretized over the wind farm area and as functions of wind direction. For the spatial discretization, a north-oriented regular mesh of $4\times6$ flow correction nodes was superimposed to the farm, as shown in Fig. 1b. Given the expected smoothness and relatively large scale of the intra-plant flow features, the spacing of the nodes is several times larger than for the Sedini case, with node-to-node distances of 4 km in the eastern direction and 4.4 km in the northern
one. A grid size convergence study showed this spacing to be dense enough to properly resolve the relevant flow features, while coarse enough for a reasonably fast computing time. Just like for Sedini, wind direction variability was taken into account by using a different spatial set of parameters every $30°$, for $\Gamma_0 \in [0°, 30°, \ldots, 330°]$. These parameters could have been further discretized in terms of atmospheric stability. However, for practical and computational reasons, the identification was instead performed twice, once considering only stable conditions, and once using only the unstable
data points. As a result, each problem was defined by 24 spatially distributed speedup parameters $\mathbf{p}_U$ for 12 different wind directions, resulting in a total of 288 to-be-identified parameters.

– For the wind direction, a single parameter $p_\Gamma$ was chosen, in order to account for any global offset in the wind direction. In fact, heterogeneous wind direction fields over the farm domain were not observable in the available dataset.





    – The wake model tuning parameters $\mathbf{p}_W$ were chosen as in the Sedini case, with the exception of the near-wake parameters

680        $\alpha$ and $\beta$, which were omitted on account of the large turbine spacing.

Based on these choices, the vector of parameters was defined as $\mathbf{p} = [\mathbf{p}_U, p_\Gamma, \mathbf{p}_W]^T$, containing a total of $N_p = 295$ free quantities. The identification was performed with three iterations of orthogonal decomposition followed by MLE. At the last iteration, 211 and 216 orthogonal parameters were retained for the unstable and stable identifications, respectively.

### 3.2.4 Coastline effects

The influence of the Danish coastline about 20 km west of the Anholt wind plant has already been analyzed by Peña et al. (2017) and van der Laan et al. (2017). In particular, the former reference investigated the speed gradients that can be observed for westerly winds, by comparing SCADA data with WRF simulations. Here, this comparison is instead performed with the heterogeneous corrections learnt by the STL method. To capture only coastline effects, it is useful to exclude plant-induced effects as much as possible from the analysis. To this end, the comparison is performed considering only the front row of

turbines for the unstable dataset, as in this case the term $\Delta U_{\mathrm{wake}\rightarrow\mathrm{amb}}$ plays a lesser role. The WRF simulation results were filtered for stability with the same criteria used for the field data, and are noted $\mathrm{WRF}_{\mathrm{unstab}}$.

    Figure 13 shows the WRF-computed speedup field generated from all situations where the wind direction is $240° \pm 5°$. Speedups were computed with respect to the average speed measured at the freestream turbines (marked in red), similarly to the STL case. The figure clearly indicates the presence of a wind speed gradient in the inflow of the wind farm, resulting from

the wake of the Jutland peninsula. For the wind rotating to the north-west, the wake of the peninsula shifts more towards the southern edge of the wind farm, whereas the opposite happens for wind rotations towards the south-west.

    Figure 14 shows the speedup factors for the western wind directions $240°$, $250°$, $260°$, and $270°$. Two sets of speedups are reported in the figure: the simulated ones, which are labelled $\mathrm{WRF}_{\mathrm{unstab}}$ and were obtained by interpolating the simulated flow field along the front row of turbines, and the identified ones, labelled $\mathrm{STL}_{\mathrm{unstab}}$ and computed by interpolating the nodal

parameters at the turbine positions and wind direction. As expected, for smaller wind direction angles, the speedups for the southern turbines are below the value of 1, indicating a decrease in inflow speed caused by the wake of the peninsula. As the wind rotates to the north, the wake of the peninsula shifts to the south; eventually, for $270°$, only the southernmost turbines are affected.

    The speedup factors from $\mathrm{WRF}_{\mathrm{unstab}}$ and $\mathrm{STL}_{\mathrm{unstab}}$ are generally in a good agreement. The remaining discrepancies can

be explained as follows:

    – The speedup factors for $\mathrm{STL}_{\mathrm{unstab}}$ were identified for discrete wind directions at intervals of $30°$. For a given wind direction, the speedup was then linearly interpolated from the identified discrete values. On the other hand, SCADA measurements were binned at intervals of $10°$, which means that the wind direction dependencies of the two sets of results have different resolutions.

– WRF simulations were run without the presence of the turbines, which therefore cannot include plant-induced effects. On the other hand, STL results are based on measured turbine data, which automatically includes such effects.

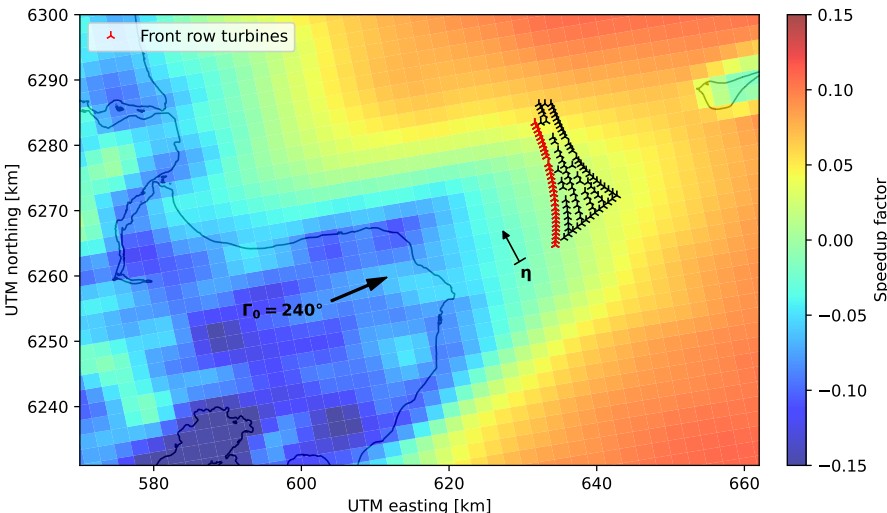

**Figure 13.** WRF-computed (Peña et al., 2017) wind speedup field in proximity of the Anholt site for wind directions $240° \pm 5°$ at hub height, in unstable conditions, without considering the wind turbines. The speedup factors are referred to the front row average. The coordinate direction $\eta$ is always perpendicular to the wind direction, originating at turbine A01.

These results show that the STL method is capable of detecting the inflow heterogeneity at this site. A similar capability was achieved – albeit in a less general setting than here – by Schreiber et al. (2019), introducing an ad hoc correction field to the inflow of the FLORIS model.

### 3.2.5  Plant-induced effects

Plant-induced flow effects account for various complex, often interrelated, phenomena. At a macroscopic level, a wind farm acts similarly to a local patch of increased surface roughness in its encounter with the atmospheric flow, leading to the development of an internal boundary layer (Porté-Agel et al., 2020). Additionally, wind tends to be "blocked", i.e. to flow around a farm rather than through it, especially in stable atmospheric conditions; this may lead to significant speed drops within the boundaries of the plant, with consequent power losses (Bleeg et al., 2018). At the same time, as the flow turns around the obstacle represented by the farm, it may locally accelerate close to its edges, resulting in increased local power outputs (Mitraszewski et al., 2013). In stable conditions, wind plants can act similarly to large orographic features such as hills and mountains, resulting in the generation of gravity waves (Teixeira, 2014). This phenomenon produces pressure changes in front and within the wind plant, which can locally negatively or positively affect power capture (Smith, 2010). Individual wind turbines are also responsible for local blockage effects: in fact, in regular arrays, the flow can be channeled in between adjacent rows of turbines, resulting in local streaks of accelerated flow (Abkar and Porté-Agel, 2013).

WIND
ENERGY
SCIENCE
DISCUSSIONS

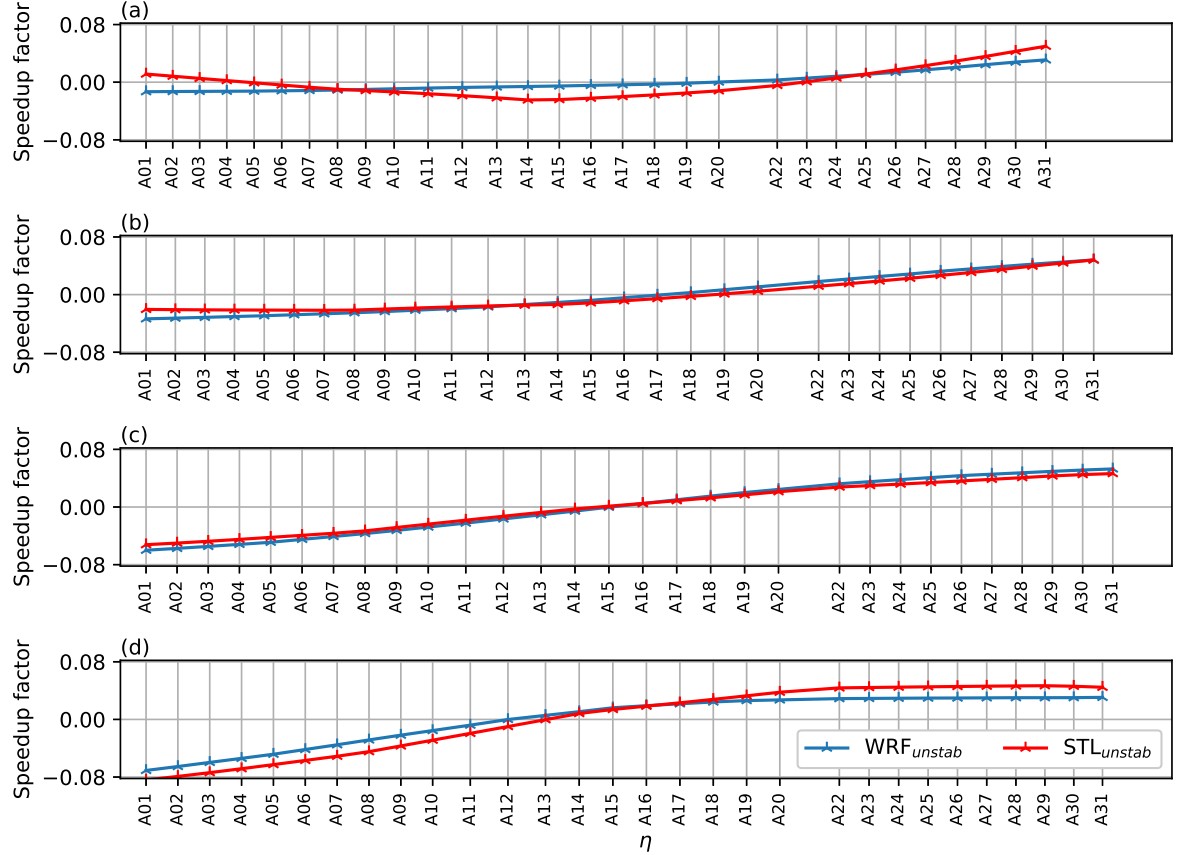

**Figure 14.** Speedup factors at the first row of turbines for westerly winds from $240°$ (**a**), $250°$ (**b**), $260°$ (**c**), and $270°$ (**d**). The coordinate $\eta$ is shown in Fig. 13, and the tick positions are proportional to the lateral distance between the turbines, normal to the wind direction.

The present method employs a correction term $\Delta U_{\mathrm{wake}\rightarrow\mathrm{amb}}$ to model any flow heterogeneity, irrespectively of its originating phenomenon. Such an approach is general and capable of very significantly boosting the quality of the match of the flow model with actual measurements. On the other hand, the drawback is that it may be difficult to disentangle one effect from the other. For example, blockage or gravity waves – when present – will affect the flow and the production of the turbines that, in turn, will generate a corresponding background flow in the model that captures such effects. However, looking at the identified flow field, it might not be possible to recognize blockage per se, as one would also need measurements of the wind speed in front of the plant, for example from a lidar, a met mast, or even an isolated upstream wind turbine. Likewise, detecting the presence of gravity waves might not be possible using only data coming from the turbines, because they operate below these higher altitude phenomena. In general, multiple concurrent phenomena can be disentangled only if they are driven by different sets of parameters, and if one has access to datasets that contain the necessary variability of such parameters. These two conditions might not always be met, and in fact they are not in the present case.



The ability to explain the results of data-driven approaches remains a topic of central importance for future research. A possible way to address this need is to resort once again to a grey-box approach, by embedding within FLORIS additional models for blockage, local accelerations, gravity waves and other effects, and tuning their parameters based on data, similarly to what is done here for the wake models. The estimated background flow would at that point represent corrections to those models, in charge of accounting for their deficiencies and any missing physics. This possible extension of the present formulation is not considered further, and the present study is limited to the identification of a "catch-all" correction term, without the pretence of being able to fully explain what has been identified. Although the explanation of this term might not be complete, it is still capable of correcting the baseline FLORIS model, substantially improving its match with respect to the measurements.

Notwithstanding these limitations of the present study, an effort was made to pragmatically separate some effects as much as possible. Specifically, orography-induced effects were reduced by considering northern and southern wind directions, where the influence of the neighboring coastlines is minimal. Conveniently, for these wind directions, the Anholt wind plant presents a significant streamwise depth, which facilitates the onset of deep-array effects. This agrees with the findings of Doekemeijer et al. (2022), who – using a uniform background flow with the baseline FLORIS model – reported an increased model mismatch from northern and southern directions.

Figure 15 reports the identified speedup fields for the stable and unstable data subsets. These results suggest the following observations.

First, the wind speed fields that are identified for stable conditions deviate significantly from those obtained in unstable conditions. In fact, the flow field seems to have a higher degree of heterogeneity in stable conditions and, as expected, intra-plant effects appear to be generally more pronounced.

Second, speedups at the edges of the wind farm can be observed for the directions $0°$, $30°$, $180°$, and $210°$. In all these cases, the flow appears to be locally accelerating, while turning around the obstacle represented by the plant.

Third, there seems to be a streamwise velocity decrease in the background flow field, especially for stable conditions, indicating the growth of a fully developed flow region. The higher mixing promoted by unstable atmospheric conditions probably induces an entrainment of the higher speed that flows over and around the array (Porté-Agel et al., 2020), reducing the streamwise deceleration. Due to the lack of models for comparison, this explanation remains of a speculative nature. Models for fully developed wind farm flows (Frandsen et al., 2006) assume a regular layout, which is not the case for the Anholt wind plant. Furthermore, they do not predict the onset distance of a possible deep-array zone.

An additional problem with the interpretation of the results is due to the fact that the identified flow correction can be affected by the wake combination scheme. As the number of wake overlaps grows towards the trailing edge of the farm, any inaccuracy in the combination model will be amplified there. The results of the figure were obtained with the SOSFS model, which has been reported to overpredict power deep inside the farm (Hamilton et al., 2020). To verify the sensitivity of the solution to this sub-model, the identification was repeated with the FLS combination, which has been reported to show a lower accuracy in the entrance region where few wakes are present, but a better power estimation in the deep part of the farm (Hamilton et al., 2020). With the FLS model the correction patterns had similar shapes, but a less pronounced magnitude in the farm deep region. Both methods are based on physical arguments and empirical assumptions, and there is no general agreement on whether one is



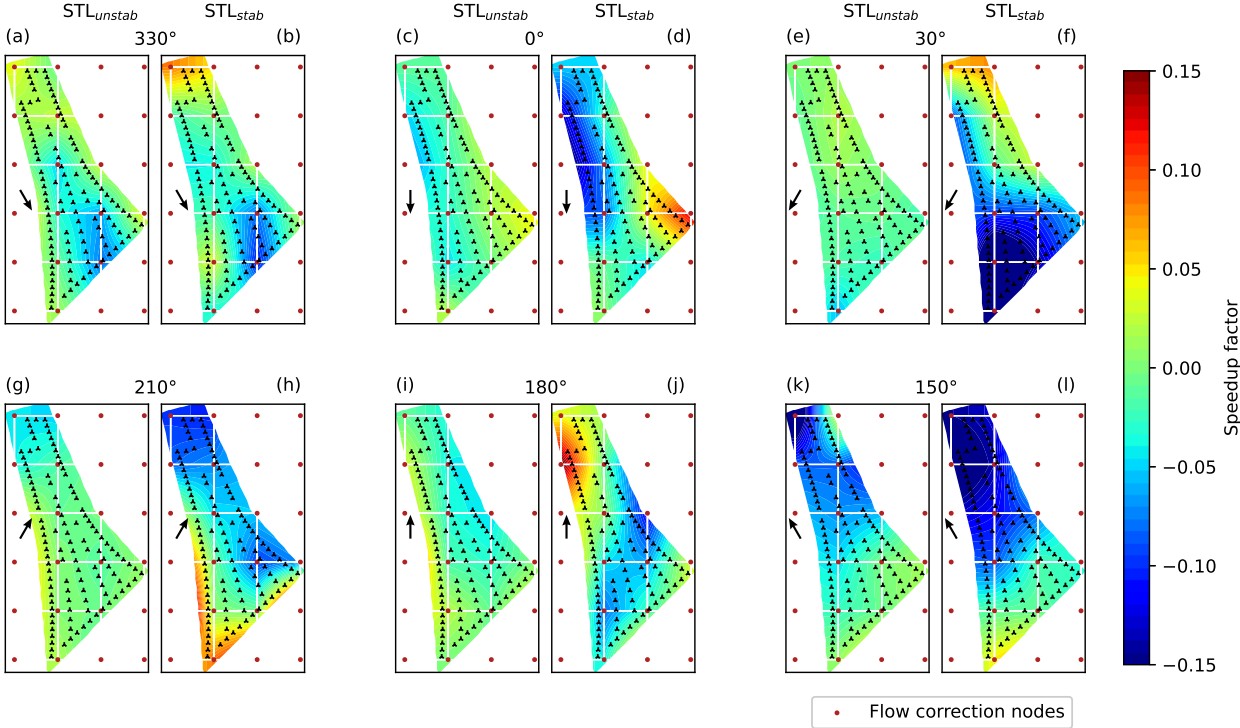

**Figure 15.** Learnt speedup fields for various wind directions, for STL$_\text{unstab}$ (unstable conditions) and STL$_\text{stab}$ (stable conditions), showing different large-scale wind farm effects. The directions $0°$, $30°$, $180°$, and $210°$ exhibit clear edge/wall effects, where the flow locally accelerates while turning around the farm.

superior to the other. Unfortunately, a correlation check through the STL orthogonal components, as performed in §3.1.4 for the wake model, is not possible in this case. In fact, these combination models do not have tunable parameters; additionally,

tuning the wake models affects all turbines, including the ones with no or limited wake overlaps in the farm entrance region.

These results highlight a problem that deserves attention and further research. In fact, the approach of adding a background correction term to the FLORIS model is somewhat oblivious to the deficiencies of its submodels: for each different wake combination model, a different background flow field is identified that, in the end, is capable of delivering a similar good match of the power predictions with the measurements, compensating possible differences in the behavior of the models.

While on the one hand this "obliviousness" is one of the strengths of learning-based data-driven methods, on the other hand it is clearly also one of their main weaknesses, because it tends to mask possible problems of the submodels, hindering a full understanding of their true capabilities.



### 3.2.6  Contributors to the error improvement

Stability affects not only the identified background flow, but also the simultaneous tuning of the wake model parameters. For the
stable and unstable cases, Table 4 lists the identified wake parameters, and compares them to their default values (NREL, 2021).
Based on the tuned model parameters, a fully waked turbine at 7 D distance produces 1% more power in unstable conditions
and 7% in stable conditions, when compared to the standard tuning. As previously mentioned, typical values for turbulence
intensity were based on the Horns Rev farm, because the actual values at the Anholt site were not available. Therefore, these
different calibrations could be due to the STL algorithm adjusting the model to the assigned values of ambient turbulence.

**Table 4.** Results of the wake model tuning, with the initial baseline parameters $k_{\mathrm{init}}$, the identified values of the additive corrections $p_k$,
and the final tuned parameters $k$. The last row reports the relative change from $k_{\mathrm{init}}$ to $k$. The left subcolumn reports values for $\mathrm{STL_{unstab}}$,
whereas the right values for $\mathrm{STL_{stab}}$.

| | $k_a$ | | $k_b$ | | $I_{\mathrm{constant}}$ | | $I_{\mathrm{ai}}$ | | $I_{\mathrm{initial}}$ | | $I_{\mathrm{downstream}}$ | |
|---|---|---|---|---|---|---|---|---|---|---|---|---|
| $k_{\mathrm{init}}$ | 0.38 | | 0.004 | | 0.8 | | 0.73 | | -0.32 | | 0.0325 | |
| $p_k$ | 0.02 | 0.08 | -0.001 | 0.0 | 0.3 | 0.0 | 0.38 | 0.05 | -0.05 | -0.06 | 0.0040 | -0.0001 |
| $k$ | 0.40 | 0.46 | 0.003 | 0.004 | 1.1 | 0.8 | 1.11 | 0.78 | -0.37 | -0.38 | 0.0365 | 0.0324 |
| $\pm$ | 5% | 21% | -27% | - | 36% | - | 52% | 7% | -16% | -19% | 12% | -1% |

Next, the performance of the STL method was compared to the baseline untuned homogeneous-background case, consid-
ering the successive activation of the various correction terms. Figure 16a shows the reduction of the root mean square error
defined by Eq. (30), suggesting a few interesting observations:

–  The initial error for the baseline model is higher in the stable case than in the unstable one. This is to be expected, for
   wake and farm effects are more prominent in stable atmospheric conditions.

–  The flow correction term produces, similarly to the Sedini case, the largest contribution to the improvement of the error.
   As for Sedini, even here this term contains clear land-induced effects, generated by the neighboring coastline. However,
   even more prominent effects are driven by the growth of the boundary layer over the farm, because of its relatively large
   streamwise depth.

–  The wake model correction is much larger than in the Sedini case. This is also to be expected, as wake interactions play
a larger role in this deeper farm.

–  In contrast to the Sedini case, the wind direction correction $p_\Gamma$ plays only a very minor role here. This is due to the better
   quality of the yaw signals, which leads to more reliable wind direction estimates that do not suffer from a large bias.

Figures 16b,c show the error probability distributions for the various methods, and for two wind speed regimes. Looking at
the baseline cases with no parameter tuning, the error has a wide spread, and the flow model is mostly over-predicting wind

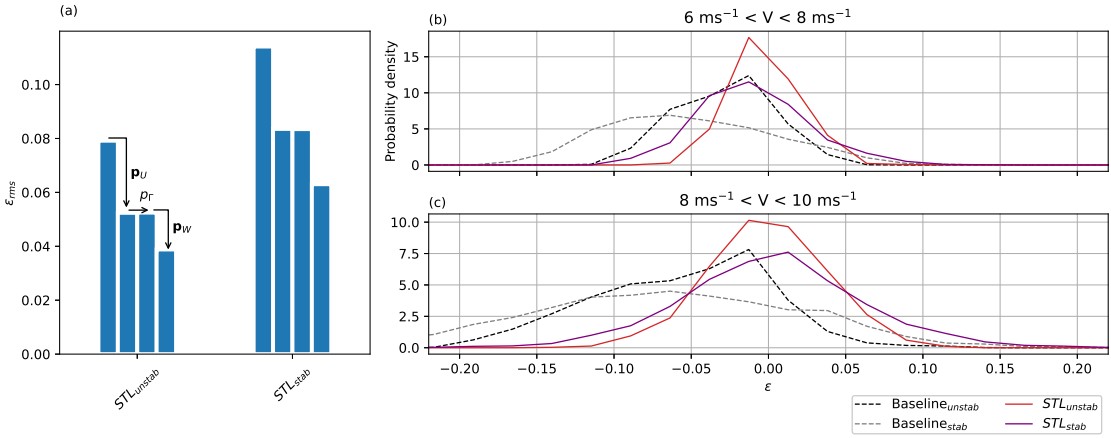

**Figure 16.** Reduction of the overall error by the activation of different correction types (**a**). Error probability density distribution for different wind speed ranges (**b**, **c**).

farm power. Both in the stable and unstable conditions, tuning and learning were able to eliminate over-predictions, reducing the spread and centering the distributions around zero.

# 4 Conclusions

The present paper has formulated and demonstrated the STL method, which simultaneously calibrates and augments a steady-state parametric wind farm flow model; this work extends an earlier less general formulation first described in Schreiber

et al. (2019). The approach builds on the vast body of knowledge and experience embedded in available engineering wake models. However, it also acknowledges that any such model will always have limited predictive accuracy because of modeling approximations and missing physics. To correct for these deficiencies, a hybrid data-driven strategy is used in the STL approach, where the baseline (white) model is augmented with ad-hoc (black-box) extra correction terms. Operational data from the farm is then used to concurrently tune the parameters of the white model, and estimate the ones of the black one.

A decomposition of the wind farm flow field by temporal and causal effects forms the basis for the definition of the extra correction terms, together with their functional dependencies and assumed parametric discretizations. The formulation allows for the first time to learn a two-dimensional heterogeneous background flow directly from operational data. In this sense, the whole wind farm is used as a distributed sensor, which detects the development of the flow within its own boundaries through the response of its wind turbines (which act as local flow sensors). The learnt heterogeneous flow is influenced by ambient

conditions, terrain orography, roughness, sea state and plant-induced effects. The learnt corrections are not limited to wind speed, but can also include heterogeneous wind direction or turbulence intensity fields.

Tuning and learning result in a severely ill-posed identification problem, because of collinearity and/or lack of observability of the redundant unknown parameters. This problem is solved by an SVD-supported MLE. The SVD effectively performs a





generalized modal decomposition of the whole solution, which includes the coupled effects of the heterogeneous flow field and
of the other tunable model parameters. This way, combinations of the parameters that are not visible – given the necessarily
limited informational content of the available dataset – can be readily discarded, whereas only visible combinations are retained.
As a byproduct of this analysis, the examination of the underlying coordinate transformation and resulting mode shapes can be
used to reveal interesting features of the solution.

The methodology was showcased via two distinct applications.

For the onshore Sedini farm, the STL revealed the existence of an heterogeneous wind speed field. Augmenting the baseline
model with this learnt background correction, together with the site-specific tuning of the wake model, resulted in a very
significant improvement in the prediction of power output throughout the farm, even when compared to the predictions of
the ad-hoc tuned baseline model. The learnt corrections showed a significant correlation with the terrain elevation, suggesting
that the observed heterogeneity of the flow is primarily driven by orographic features of the site. This was further confirmed
by over-the-terrain CFD simulations, which also showed a good agreement with the learnt corrections. Additionally, the CFD-
computed flow field was used as an initial starting guess for the learnt correction term; this however did not significantly change
the results. Furthermore, the STL was able to identify a large bias in the wind direction, presumably due to problems with the
wind turbine yaw encoders.

For the much larger offshore Anholt farm, the STL revealed the existence of gradients in the inflow, as well as the presence
of a strongly direction and stability-dependent highly heterogeneous intra-plant flow field. Comparison with WRF simulations
confirmed the origin of the inflow gradients as caused by the presence of coastlines in close proximity of the farm, as already
observed by other authors. The intra-plant flow exhibited clear instances of local accelerations close to the farm edges, sug-
gesting that the flow is "turning" around the obstacle represented by the farm. The observed intra-plant flow appears to be
caused by the growth of the boundary layer over the farm. The flow appears to be very significantly influenced by the irregular
shape of the farm and by the spacing of the turbines, which would be difficult to capture with simplified analytical models.
However, the interpretation of the results was complicated by the effects caused by the interaction of multiple wakes towards
the farm trailing edge. It was in fact observed that changes to the wake combination model can affect the identified background
flow. Given the present dataset, it was not possible to disentangle the two effects, which remains an open problem that will
necessitate further research. Notwithstanding this limitation, the STL was able to very significantly improve the prediction of
power when compared to the tuned baseline, no matter what wake combination model was used.

Future work can further improve the STL approach.

On the white-box side of the problem, it would be interesting to add the most recent generation of intra-plant effects. This
could help disentangle the causes for the observed heterogeneous background corrections in large farms. Similarly, one should
explore the use of more sophisticated wake combination models than the ones used here, given their significant effects on
the estimated background flow. Models that are parametric (i.e., that can be tuned) would be of particular interest, given the
"monolithic" parameter estimation performed by the STL.

On the black-box side of the problem, the use of richer datasets than the ones used here could really help illuminate some of
the complexities of wind farm flows. For example, operational data accompanied with information on the ambient conditions





could help better discern the effects of stability on phenomena such as boundary layer growth, blockage, gravity waves, and
others. Additionally, extra measurements provided on site by met masts and/or long-range scanning lidars could be fused with
the operational data, boosting the informational content of the dataset. Although the grey-box nature of the STL method means
that the white box component can compensate for lack of information in the data, it is also true that what is not in the white box
and not in the data can never be correctly represented by the model. Therefore, future improvements depend to some extent on
the richness and quality of the datasets that will be available.

Finally, the STL method should be extended to incorporate unsteady effects, by the use of a dynamic version of the baseline
engineering wake model. It is envisioned that the steady-state STL could be used, as done here, to adapt the model to represent
permanent features of the flow (for example, as caused by a hill), whereas the unsteady STL could be used to render any
transient effects (for example, as caused by the finite-speed propagation downstream of set point or inflow changes).

**Appendix A: CFD simulations of the flow over the terrain**

For the Sedini case, RANS simulations were carried out in OpenFOAM (v2006) in neutral atmospheric conditions, with the
goals of generating a term of comparison for the learnt flow corrections (see § 3.1.5), and of providing a non-uniform initial
guess to the STL algorithm (§ 3.1.6).

    The learnt corrections $\Delta U_{\mathrm{STL}}$ appeared to be independent of the inflow wind speed. In accordance with common practice
in industrial applications (van der Laan et al., 2020), the same Reynolds independence was observed for the CFD simulations.
The relative variation of wind speed at hub height was computed as

$$\Delta \hat{U}_{\mathrm{CFD}}(\mathbf{A}_0, Q) = \frac{U_{\mathrm{CFD}} - U_{0,\mathrm{CFD}}}{U_{0,\mathrm{CFD}}} \ . \tag{A1}$$

The reference speed $U_{0,\mathrm{CFD}}$ was obtained as the average at the free-stream turbine positions, similarly to the field data case.
Due to the Reynolds independence and assumption of neutral conditions, wind direction is the only environmental dependency
that was considered, i.e. $\mathbf{A}_0 = \Gamma_0$. One simulation was run for each of the directions $\Gamma_0 \in [255°, 270°, 285°, 300°]$.

**A1   Domain and mesh generation**

A rectangular domain was used, because of its simpler mesh generation and clear identification of inlet/outlet compared to other
shapes. For each different wind direction case, the terrain was rotated to align the domain with the inflow. Figure A1a shows
the domain boundaries and the 2×3 km farm located at its center for the 270° case. The terrain was modeled with satellite
DEM data (ASTER, 2021), with a 55 m resolution. A smoothing kernel was applied to the terrain to promote the progressive
growth of a boundary layer downstream of the inlet. This was obtained by modifying the terrain elevation (Sørensen et al.,
2012) with the following function

$$f_k = \tanh\left[\left(\frac{r}{R}\right)^6\right] , \tag{A2}$$





where $R = 6$ km is the radius where the terrain elevation vanishes, obtaining the smoothed out domain shown in Fig. A1b. A domain height of only 3 km was found to be sufficient to ensure an undisturbed flow at the top of the domain, although this
value is smaller than the one recommended by Sørensen et al. (2012).

(a)                                                                (b)

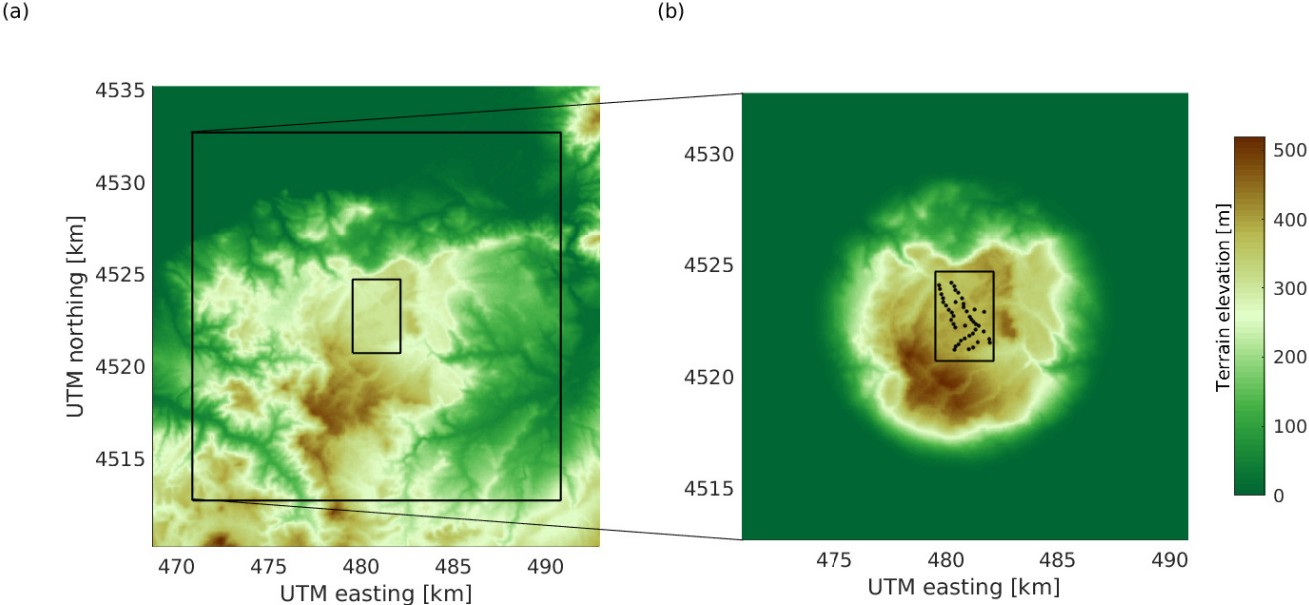

**Figure A1.** Terrain elevation around the Sedini wind farm (**a**). The large rectangle shows the CFD domain, whereas the small one shows the perimeter of the wind farm (visible in the right panel). Terrain elevation after application of the smoothing kernel (**b**). The figures correspond to the 270° wind direction case; for other directions, the computational domain was rotated accordingly.

A regular background mesh was generated with the blockMesh tool that is part of the OpenFOAM distribution. In the horizontal direction, $N_x = N_y = 500$ cells were used, resulting in a resolution $\Delta x = \Delta y = 40$ m. In the vertical direction, the domain was divided in two parts at a height of 1 km. The top section had a constant vertical spacing $\Delta z_{\text{top}} = 100$ m. In the bottom section, a grading scheme was used to progressively increase the spacing from $\Delta z_{\text{bott}} = 5$ m close to the ground to
25 m at the split section. An additional layer with a thickness of 2.7 m was added to further refine the region close to the ground, resulting in an average aspect ratio of the wall-adjacent cells equal to 14.8. Finally, the tool SnappyHexMesh was used to adapt the $\approx 16.5$ million-cell-grid to the terrain contour.

## A2   Boundary conditions and numerical setup

As simulations were performed only for neutral conditions, buoyancy effects were not included. Furthermore, Coriolis effects
were also neglected as only the velocity at hub height is of interest, which is very close to the surface. The k-$\epsilon$ model was used





for turbulent stresses. As suggested by Richards and Hoxey (1993), the model constant $\sigma_\epsilon$ was set to the value of 1.11, which is typical of atmospheric flows, while the other model parameters were left at their default values (Launder and Spalding, 1974).

The domain boundary conditions were imposed as follows. A logarithmic velocity profile was imposed at the inlet, with a roughness length $z_0 = 0.01$ – corresponding to open terrain, as the site has not much vegetation – and hub-height speed $U_{\mathrm{hh}} = 8$ ms$^{-1}$. The inlet profiles for turbulent kinetic energy $k_{\mathrm{turb}}$, eddy viscosity $\nu_T$ and dissipation of turbulent kinetic energy $\epsilon_{\mathrm{turb}}$ were implemented with the ABL boundary conditions of OpenFOAM version v2006 (Release Notes v2006, 2020), based on Hargreaves and Wright (2007) and Yang et al. (2009). No-slip conditions and wall functions based on Hargreaves and Wright (2007) were used in proximity of the terrain. At the top of the domain, a constant shear stress was used to drive the flow, while symmetry conditions were applied to the side walls. At the outlet, the kinematic pressure was set to zero, while zero gradient conditions were imposed for all other variables.

A second-order accurate, linear discretization scheme was used for the divergence terms. The problem was solved with simpleFOAM, an implementation of the SIMPLE algorithm. The setup was first tested in an empty domain, where it was able to establish an equilibrium ABL with constant velocity profile from inlet to outlet. The four simulations were run with 322 cores and converged after ca. 1200 iterations.

## A3 Grid convergence

To investigate grid convergence, the mesh was progressively coarsened in both the horizontal and vertical directions (in the latter case, only in the bottom section of the domain), obtaining 10%, 30%, and 50% fewer grid points, respectively. Table A1 lists the defining parameters for the fine and the coarser domains for the 270° direction; similar results were obtained for the other directions. The average cell height in the bottom section of the domain is noted $\langle \Delta z_{\mathrm{bott}} \rangle$.

**Table A1.** Mesh characteristics for the grid convergence study for the 270° direction.

| Grid Level | $N_x$, $N_y$ | $\Delta x, \Delta y$ [m] | $N_z$ (top/bottom) | $\langle \Delta z_{\mathrm{bott}} \rangle$ [m] | $N_{\mathrm{grid}}$ [$10^6$] | $\langle \epsilon_{\mathrm{GL,i}} \rangle$ [%] | $\max(\epsilon_{\mathrm{GL,i}})$ [%] | $\mathrm{std}(\epsilon_{\mathrm{GL,i}})$ [%] |
|---|---|---|---|---|---|---|---|---|
| Fine | 500 | 40 | 80/20 | 12.5 | 16.4 | - | - | - |
| Coarse 1 | 450 | 44.5 | 72/20 | 13.3 | 12.0 | 0.17 | 1.6 | 0.33 |
| Coarse 2 | 350 | 57 | 56/20 | 17.8 | 5.9 | 0.62 | 2.3 | 0.45 |
| Coarse 3 | 250 | 80 | 40/20 | 25 | 2.3 | 1.17 | 4.0 | 0.74 |

Figure A2 shows the average hub-height speed difference $\langle \epsilon_{\mathrm{GL,i}} \rangle$ for the three coarser cases with respect to the fine-grid solution, where $\epsilon_{\mathrm{GL,i}} = (U_{\mathrm{GL,base}} - U_{\mathrm{GL,i}})/U_{\mathrm{GL,base}}$. The coarser solution, based on 50% fewer grid points than the fine one, differs only by about 4%. The curve trend indicates that the fine-grid solution can be considered to be at convergence.



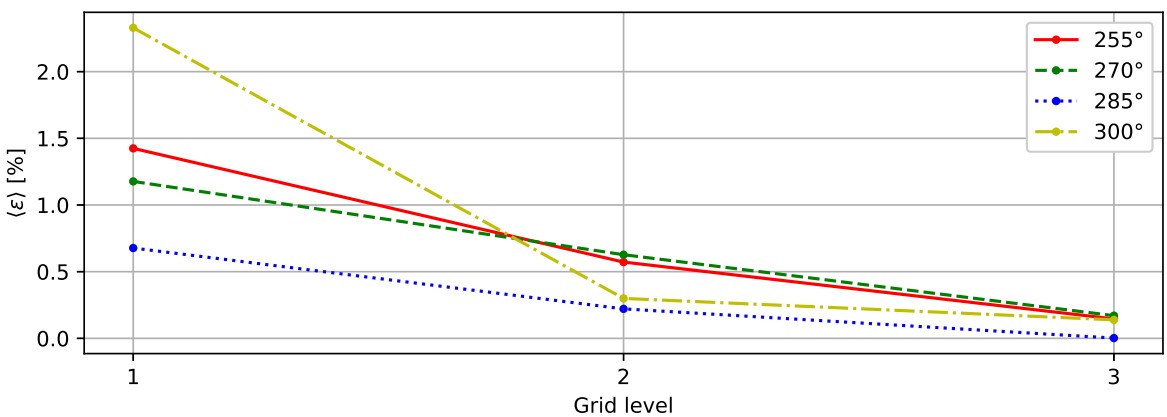

**Figure A2.** Average relative difference of hub-height speed for grids of increasing coarseness.





## Appendix B: Results for the Sedini wind plant during nighttime operation



**Figure B1.** Normalized measured and calculated power, for all $5°$ bins in the investigated $245°$-$310°$ sector, for wind speeds in the range $6$-$8$ ms$^{-1}$, during nighttime operation. Bins with $< 10$ observations are not shown. The uncertainty band indicates the standard deviation in the bins. $\Delta h$ is the foundation elevation difference with respect to the farm average.





**Appendix C: Nomenclature**

| | | |
|---|---|---|
| 925 | $\mathbf{A}$ | Vector of ambient state variables |
| | $\mathbf{E}$ | Fisher matrix |
| | $F$ | Flow quantity |
| | $h$ | Elevation |
| | $I$ | Turbulence intensity, |
| 930 | $J$ | Cost function |
| | $k$ | Wake model parameter |
| | $\mathrm{L}$ | Obukhov length |
| | $\mathrm{M}$ | Sensitivity matrix |
| | $\mathbf{n}$ | Shape function vector |
| 935 | $N$ | Number of observations |
| | $N_t$ | Number of turbines |
| | $N_p$ | Number of STL parameters |
| | $N_r$ | Number of retained orthogonal parameters |
| | $\mathbf{p}$ | Complete vector of free STL correction parameters |
| 940 | $\mathbf{p}_F$ | Correction node values to model flow heterogeneity |
| | $\mathbf{p}_W$ | Correction parameters for wake model tuning |
| | $P$ | Turbine power |
| | $\mathbf{P}$ | Inverse of the Fisher matrix |
| | $Q$ | Spatial position |
| 945 | $r$ | Residual between measurement and model output |
| | $\mathbf{R}$ | Measurement covariance matrix |
| | $s$ | Singular value |
| | $\mathbf{S}$ | Matrix of singular values |
| | $U$ | Wind speed at hub height |
| 950 | $\mathbf{U}$ | Matrix of left singular vectors |
| | $\mathbf{v}$ | Right singular vector |
| | $\mathbf{V}$ | Matrix of right singular vectors |
| | $w$ | Measurement weight |
| | $y$ | Model output |
| 955 | $z$ | Measurement |
| | $z_0$ | Roughness length |



| | |
|---|---|
| $\Gamma$ | Wind direction |
| $\Delta F$ | Heterogeneous flow quantity correction |
| $\epsilon$ | Error |
| $\theta$ | Orthogonal parameter |
| $\sigma_m$ | Measurement variance |
| $\sigma_t$ | Observability threshold |
| $\boldsymbol{\Psi}$ | Matrix of eigenshapes |
| | |
| $\bar{(.)}$ | Constant-in-time component |
| $\tilde{(.)}$ | Slow component |
| $(.)'$ | Turbulent (fast) component |
| $(.)_0$ | Site-average quantity |
| $\hat{(.)}$ | Scaled quantity |
| $\langle . \rangle$ | Average operator |
| | |
| ABL | Atmospheric boundary layer |
| FLORIS | FLOw Redirection and Induction in Steady State |
| FLS | Freestream linear superposition |
| LIDAR | Light detection and ranging |
| MLE | Maximum likelihood estimation |
| RANS | Reynolds-averaged Navier-Stokes |
| REWS | Rotor-equivalent wind speed |
| SCADA | Supervisory control and data acquisition |
| SOSFS | Sum of squared freestream superposition |
| STL | Simultaneous tuning and learning |
| SVD | Singular value decomposition |
| TI | Turbulence intensity |
| WRF | Weather research and forecasting model |

*Code and data availability.* Data of the Sedini wind farm is the property of Enel Green Power S.p.A. Data of the Anholt wind farm is the property of Ørsted A/S. MATLAB figure files that allow for the lossless extraction of results can be retrieved via the DOI XXXX[1]. A Python implementation of the STL method can be provided upon request by contacting the corresponding author.

---

[1]Review note: the DOI will be generated if the paper is accepted.





*Author contributions.* CLB proposed the method of the wind farm as a sensor, formulated the STL algorithm, and supervised the research. RB and AV implemented the method; AV developed the data processing procedures. RB developed the application to the Sedini wind farm, and AV to the Anholt plant. RB developed the over-the-terrain CFD approach, and performed the numerical simulations. All authors equally contributed to the interpretation of the results. RB and CLB wrote the manuscript, with contributions by AV in the Anholt section. All authors provided important input to this research work through discussions and feedback and by improving the manuscript.

*Competing interests.* The contact author has declared that none of the authors has any competing interests.

*Acknowledgements.* The authors express their gratitude to Enel Green Power S.p.A. and Ørsted A/S, which granted access to the Sedini and Anholt field data, respectively, and to Achim Fischer for his advice on over-the-terrain CFD.

*Financial support.* This work is funded in part by the e-TWINS project (FKZ:03EI6020A), which receives funding from the German Federal Ministry for Economic Affairs and Climate Action (BMWK).



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
