# Peer review of "The wind farm as a sensor: learning and explaining orographic and plant-induced flow heterogeneities from operational data"

_Wind Energy Science, 2022_

## Referee Comment (RC1)

**Review of the paper "The wind farm as a sensor: learning and explaining orographic and plant-induced flow heterogeneities from operational data" by R. Braunbehrens, A. Vad and C. L. Bottasso with MS No. wes-2022-67**

3 November 2022

**General comments:**

This paper describes a method to identify the heterogenous flow characteristics that develop within a wind farm in its interaction with the atmospheric boundary layer. The proposed method is based on augmenting an engineering model with an unknown correction field. Operational data is used to simultaneously learn the parameters that describe the correction field, and tune the ones of the engineering model. This approach is demonstrated on a mid-size onshore farm and a large offshore one. In both cases, the data-driven correction and tuning of the proposed model results in much improved prediction capabilities. The paper is generally well written and referenced, and represents a substantial contribution to scientific progress within the scope of Wind Energy Science (WES). Thus, I am very happy to recommend the paper for publication in WES.

**Specific comments:**

1. Line 295. It would be helpful to mention that the residuals $r$ with covariance $R$ is statistically independent within the set of $N$ observations $\{z_1, z_2, \ldots, z_N\}$.

2. Equation 19. Could the authors explain somewhat how to get the values of the rotation matrix $V$ and the diagonal matrix $S$ when the values of the matrix $M$ are already known?

3. Figures 3, 4, 5. Is the abscissa really the value of the new parameters $\theta_i$? Or should it be the index of the new parameters?

**Technical corrections:**

1. Lines 45-46: Remove one "only", and "pant" should be "plant".

2. Line 80. "frestream" should be "freestream".

3. Line 101. "it" should be "is".

4. Figure 8. The title misses "(c)".

5. Line 701. "1" should be "0".

---

## Referee Comment (RC2)

**Review:** *The wind farm as a sensor: learning and explaining orographic and plant-induced flow heterogeneities from operational data*

**Summary**

The authors describe a method for identifying and reconstructing flow heterogeneity within a wind farm caused by orography and the interaction of turbine wakes with the atmospheric boundary layer. The proposed model combines measurements and an engineering wake model to recreate flow heterogeneity within a wind farm. Their model incorporates parametric corrections to a simplified, physics-based engineering wake model to represent unmodeled physics. Parametric corrections are based on atmospheric variables (e.g., wind speed, wind direction, turbulence intensity, surface stability) and the engineering wake model. They test their model in two very different wind farms: one onshore and one offshore. For the onshore wind farm, they find terrain is likely the dominant factor influencing flow heterogeneity. For the offshore wind farm, their model captured coastline effects, local speedups around the wind farm, and deep array effects. For both wind farms, the proposed model reduces the error in the power production estimate compared to the stand-alone engineering wake model. In general, the manuscript is well written, and the results support their conclusions. I recommend the paper is accepted after minor revisions.

**Specific comments**

Lines 73-74: Soften the language regarding gravity waves. Only a specific set of LES studies and some simplified linearized simulations show gravity wave initiation in very large wind farms and under a very specific set of atmospheric conditions. There is still a lot of uncertainties around this issue.

Lines 162-171: Please comment on the importance of including $\Delta U_{amb \rightarrow wake}$ . For instance, it has been shown that the wake follows the terrain in stable conditions but tends to deflect upwards in unstable conditions (e.g., Wise et al. (2022)), influencing power production of downstream turbines.

Line 326: Please explain how you define the observability threshold.

Lines 441-442: Please explain how you decided on the observability threshold. Did the choice of threshold modify the results?

Line 382 and 640: You mention rotor equivalent wind speed, which requires having the wind speed profile throughout the turbine rotor layer. Yet you are using point-measurement wind speed from a nacelle-mounted anemometer. Please make correction.

Lines 392-394: Please include distribution of TI and shear for daytime and nighttime conditions. Are TI and shear dependent on wind direction? I would expect the land/sea fetch varies by wind direction sector, modifying TI and shear.

Figure 4: Please specify in the caption what the dashed black line is.

Lines 514-515: How are you comparing the eigenshapes with terrain? Are you estimating a correlation between both? Or is it just by visual inspection?

Line 538: Although visual inspection can provide a first approximation to the agreement between both fields, a quantitative assessment is necessary given that it validates whether the proposed model captures the spatial variability. A quantitative estimate can be easily obtained by interpolating the simulation field to the learnt field from Figure 7.

Lines 539-540: Like my previous comment, you can find the correlation between terrain and speedups/slowdowns.

Figure 10: Please show the entire range of the standard deviation for panel b.

Figure 11: Given that this is a very extensive paper, consider moving this figure to the Appendix.

Lines 733-735: Gravity waves in the free atmosphere can presumably modify wind speed and pressure at hub height. Please make correction.

---

## Author Comment (AC1)

**Reply to Reviewers**

We thank the reviewers for their detailed analysis and constructive inputs. A list of point-by-point replies to the reviewers' comments is detailed in the following.

**Reviewer 1**

*This paper describes a method to identify the heterogenous flow characteristics that develop within a wind farm in its interaction with the atmospheric boundary layer. The proposed method is based on augmenting an engineering model with an unknown correction field. Operational data is used to simultaneously learn the parameters that describe the correction field, and tune the ones of the engineering model. This approach is demonstrated on a mid-size onshore farm and a large offshore one. In both cases, the data-driven correction and tuning of the proposed model results in much improved prediction capabilities. The paper is generally well written and referenced, and represents a substantial contribution to scientific progress within the scope of Wind Energy Science (WES). Thus, I am very happy to recommend the paper for publication in WES.*

1. **Reviewer**: *Line 295. It would be helpful to mention that the residuals r with covariance R is statistically independent within the set of N observations {z1, z2, ..., zN}.*
   **Authors**: We are not sure what is meant here. The residuals are in general not statistically independent, and in fact the measurement noise covariance is typically computed as $R = \frac{1}{N}\sum_{i=1}^{N} r_i\, r_i^T$, which results in a fully populated matrix (Jategaonkar, 2015).

2. **Reviewer**: *Equation 19. Could the authors explain somewhat how to get the values of the rotation matrix V and the diagonal matrix S when the values of the matrix M are already known?*
   **Authors**: The text has been expanded in this part of the paper, to improve clarity. We have also added a reference to a book on the topic (Wall et al., "Singular Value Decomposition and Principal Component Analysis"), where the reader can find more detailed information on the SVD. However, since the SVD is textbook material, we believe that a detailed explanation is not necessary.

3. **Reviewer**: *Figures 3, 4, 5. Is the abscissa really the value of the new parameters $\vartheta i$? Or should it be the index of the new parameters?*
   **Authors**: Thank you, this indeed was a bit misleading. We have now specified that the abscissa represents the parameter indices in the figure captions.

4. **Reviewer**: *Lines 45-46: Remove one "only", and "pant" should be "plant".*
   **Authors**: Thank you, corrected.

5. **Reviewer**: *Line 80. "frestream" should be "freestream".*
   **Authors**: Corrected.

6. **Reviewer**: *Line 101. "it" should be "is".*
   **Authors**: Corrected.

7. **Reviewer**: *Figure 8. The title misses "(c)".*
   **Authors**: Corrected.

8. **Reviewer**: *Line 701. "1" should be "0".*
   **Authors**: Corrected.

**Reviewer 2**

*The authors describe a method for identifying and reconstructing flow heterogeneity within a wind farm caused by orography and the interaction of turbine wakes with the atmospheric boundary layer. The proposed model combines measurements and an engineering wake model to recreate flow heterogeneity within a wind farm. Their model incorporates parametric corrections to a simplified, physics-based engineering wake model to represent unmodeled physics. Parametric corrections are based on atmospheric variables (e.g., wind speed, wind direction, turbulence intensity, surface stability) and the engineering wake model. They test their model in two very different wind farms: one onshore and one offshore. For the onshore wind farm, they find terrain is likely the dominant factor influencing flow heterogeneity. For the offshore wind farm, their model captured coastline effects, local speedups around the wind farm, and deep array effects. For both wind farms, the proposed model reduces the error in the power production estimate compared to the stand-alone engineering wake model. In general, the manuscript is well written, and the results support their conclusions. I recommend the paper is accepted after minor revisions.*

1. **Reviewer**: *Lines 73-74: Soften the language regarding gravity waves. Only a specific set of LES studies and some simplified linearized simulations show gravity wave initiation in very large wind farms and under a very specific set of atmospheric conditions. There is still a lot of uncertainties around this issue.*
   **Authors**: We agree with this remark, and the text has been updated accordingly.

2. **Reviewer**: *Lines 162-171: Please comment on the importance of including $\Delta U_{amb \to wake}$. For instance, it has been shown that the wake follows the terrain in stable conditions but tends to deflect upwards in unstable conditions (e.g., Wise et al. (2022)), influencing power production of downstream turbines.*
   **Authors**: Thank you for pointing this out. We have updated the text accordingly and added Wise et al. 2022 to the list of references.

3. **Reviewer**: *Line 326: Please explain how you define the observability threshold.*
   **Authors**: We have added an explanation on the meaning of the observability threshold.

4. **Reviewer**: *Lines 441-442: Please explain how you decided on the observability threshold. Did the choice of threshold modify the results?*
   **Authors:** We have added an analysis on the effects of changing the observability threshold, which is now shown in Fig. 3 (new in the revised manuscript). The figure shows that the choice of a value of 0.01 is behind the "knee" of the curve, and -therefore- the solution can be considered as converged. Using even higher values of the threshold would lead to only small improvements of the cost, while the computational cost would increase because of the larger number of retained parameters in the identification. On the other hand, similar results could have been obtained by reducing the threshold to about 0.03-0.04, with the advantage of a lower

computational cost. We have update the text to explain the new plot and comment on this point.

5. **Reviewer**: *Line 382 and 640: You mention rotor equivalent wind speed, which requires having the wind speed profile throughout the turbine rotor layer. Yet you are using point-measurement wind speed from a nacelle-mounted anemometer. Please make correction.*
   **Authors**: Thank you, your comment indicates that this point was unclear in the original manuscript. In fact, we did not use nacelle anemometer measurements anywhere in this study because of their typically low accuracy and of frequent gaps in the available recordings. Following Schreiber et al., 2018, the rotor equivalent wind speed was computed from the power curve and the assumed shear at the site. The text has been updated in the Sedini and Anholt sections to clarify this point.

6. **Reviewer**: *Lines 392-394: Please include distribution of TI and shear for daytime and nighttime conditions. Are TI and shear dependent on wind direction? I would expect the land/sea fetch varies by wind direction sector, modifying TI and shear.*
   **Authors**: We agree, and in fact we confirm that, depending on wind direction, TI and shear do vary at this site. However, the current study used data only from the sector 245-310°, which is almost exclusively influenced by the land inflow. A more general study considering a wider range of wind direction should take this effect into account. This was noted in the revised manuscript.

7. **Reviewer**: *Figure 4: Please specify in the caption what the dashed black line is.*
   **Authors**: The line indicates the cutoff of the orthogonal parameters. A sentence was added to the figure caption.

8. **Reviewer**: *Lines 514-515: How are you comparing the eigenshapes with terrain? Are you estimating a correlation between both? Or is it just by visual inspection?*
   **Authors**: The comparison is only qualitative, because what counts is the comparison between the terrain and the computed flow solution (please see next question). In other words, one could have a very good correlation between elevation and learnt flow field, which is computed as a superposition of multiple modes, but still a low correlation with the individual modes. Yet, seeing in the modes some features of the terrain is interesting and suggestive of the fact that the mode expansion indeed captures some terrain-induced affects, so we have left a brief discussion on this point in the text.

9. **Reviewer**: *Line 538: Although visual inspection can provide a first approximation to the agreement between both fields, a quantitative assessment is necessary given that it validates whether the proposed model captures the spatial variability. A quantitative estimate can be easily obtained by interpolating the simulation field to the learnt field from Figure 7.*
   **Authors**: Thank you for this suggestion. We have now computed correlation coefficients between CFD-simulated and learnt flow fields, and between learnt flow field and terrain elevation. The results have been presented in a new table (number 3 in the revised manuscript), and the text has been expanded accordingly. This quantitative analysis confirms and supports the conclusions from the visual inspection.

10. **Reviewer**: *Lines 539-540: Like my previous comment, you can find the correlation between terrain and speedups/slowdowns.*
    **Authors**: See previous comment.

11. **Reviewer**: *Figure 10: Please show the entire range of the standard deviation for panel b.*
    **Authors**: The y axis limits were adjusted.

12. **Reviewer**: *Figure 11: Given that this is a very extensive paper, consider moving this figure to the Appendix.*
    **Authors**: The figure has been moved to the appendix. Some small adjustments have been made to the text to reflect this change.

13. **Reviewer**: *Lines 733-735: Gravity waves in the free atmosphere can presumably modify wind speed and pressure at hub height. Please make correction.*
    **Authors**: This comment is in the spirit of remark 1 from the same reviewer, and we agree. We have adjusted the sentence, although the lack of certainty regarding the effects of gravity waves was already noted earlier in the text.

While revising the manuscript, we repeated most of the simulations and found an inconsistency: while the paper stated that we used the SOSFS method for both the Sedini and Anholt wind farms, in reality the best results could be obtained with SOSFS for Sedini and FLS for Anholt. The text has now been corrected, and figures 14, 15 and 16 have been updated accordingly. None of these changes affect the conclusions of the paper, and actually the results are even slightly better than in the previous version.

Additionally, we have taken the opportunity of this revision for making several small editorial changes to the text, in order to improve readability. We have also increased the size of the fonts of several figures, which were too small in our opinion.

A revised version of the manuscript is attached to the present reply, with the main changes highlighted in red (deletions) and blue (additions).

Best regards.
The authors

[revised manuscript text omitted]
_{\text{CFD}} = \frac{\text{cov}\left(\Delta \tilde{U}_{\text{STL}}(Q) \Delta \tilde{U}_{\text{CFD}}(Q)\right)}{\sqrt{\text{var}\left(\Delta \tilde{U}_{\text{STL}}(Q)\right) \text{var}\left(\Delta \tilde{U}_{\text{CFD}}(Q)\right)}} . \tag{32}$$

Ideally, for a perfect match between learnt and simulated fields, their spatial correlation $\varrho_{\text{CFD}}$ should be equal to one. Similarly,

590 a terrain correlation measure is defined as

$$\varrho_{\text{T}} = \frac{\text{cov}\left(\Delta \tilde{U}_{\text{STL}}(Q) h(Q)\right)}{\sqrt{\text{var}\left(\Delta \tilde{U}_{\text{STL}}(Q)\right) \text{var}(h(Q))}} , \
[revised manuscript text omitted]